# Obliviate: Efficient Unlearning in Recommender Systems

**Tushar Prakash** [1]   **Brijraj Singh** [1]   **Niranjan Pedanekar** [1]   **Narayan Chaturvedi** [2][3]

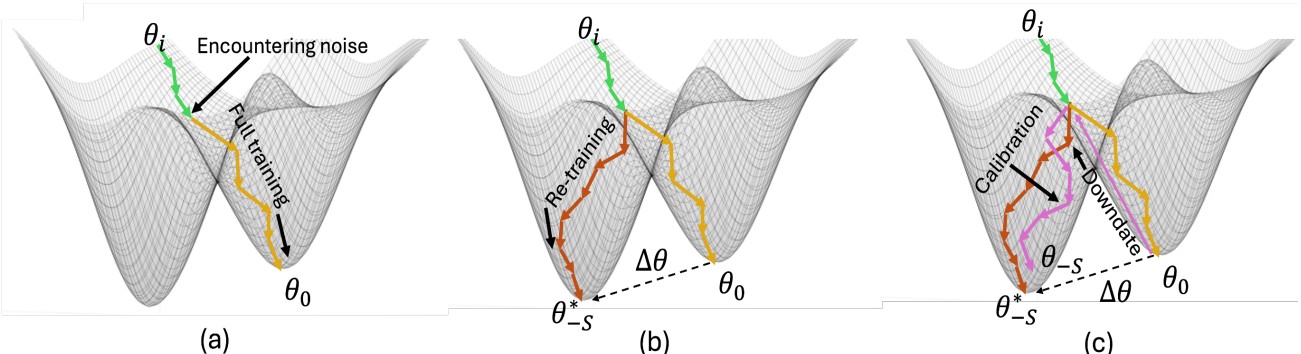

*Figure 1.* Illustration of the unlearning objective in parameter space. The yellow trajectory signifies the training on the full dataset, which also encounters deletion set (noise). The red trajectory shows a model $\theta^*_{-S}$ trained from scratch after removing the deletion set interactions. The pink trajectory shows unlearning process that aims at efficiently transforming $\theta_0$ into $\theta_{-S} \approx \theta^*_{-S}$ by estimating and removing the parameter influence $\Delta\theta$ induced by the deletion set, without full retraining.

## Abstract

Machine unlearning is becoming increasingly critical in the context of data privacy regulations, particularly for recommendation systems that are directly trained on user interaction data. The goal of this work is to remove requested interaction data and their downstream influence from trained model while preserving recommendation quality, and to do so without incurring the substantial computational cost of full retraining. Existing approaches exhibit several limitations, including limited unlearning completeness and degradation in recommendation performance, while having substantial computational overhead. In this paper, we propose **Obliviate**[1], an efficient two-stage unlearning framework for recommender systems that achieves high unlearning completeness while maintaining good utility. In the first stage, we introduce a *Low-Rank Unlearning Adapter (LUA)*, which employs a lightweight Hessian proxy to enable curvature-aware and efficient unlearning through localized low-rank adapters rather than full parameters. In the second stage, we propose *Locality-Aware Calibration (LAC)*, a lightweight refinement stage that updates only the adapter parameters to improve the performance by enforcing unlearning via ranking-based objectives while preserving utility through knowledge distillation. Extensive empirical evaluations demonstrate that **Obliviate** achieves high level of forgetting with minimal loss in recommendation quality and at significantly reduced computational cost, offering a practical and scalable solution for large-scale recommender systems.

## 1. Introduction

In the era of artificial intelligence, modern systems are trained on vast and diverse datasets to acquire broad, world-level knowledge (Liu et al., 2025; Naveed et al., 2025). At the same time, there is a growing shift toward personalization, where models adapt to individual preferences through personalized interaction data, increasing engagement with the platform (Cheng et al., 2016; Covington et al., 2016; Xia et al., 2024). It includes large-scale recommendation systems used by platforms like YouTube and Amazon, which leverage interaction histories to drive user experiences ranging from entertainment to e-commerce. Despite the sub-

[1]User Engagement Group, Sony Research India [2]Graphic Era University, India [3]Work done during tenure at Sony Research India. Correspondence to: Tushar Prakash <tushar.prakash@sony.com>.

*Proceedings of the 43rd International Conference on Machine Learning*, Seoul, South Korea. PMLR 306, 2026. Copyright 2026 by the author(s).

[1]*Obliviate* is a charm from the Harry Potter series used to erase specific memories from an individual's mind.

stantial gains in model performance and user engagement, personalization also introduces significant challenges related to data governance and user privacy, as these systems rely heavily on personal user data for training (Himeur et al., 2022; Di Fazio, 2024).

To address these concerns, regulations such as the General Data Protection Regulation (GDPR) (Mantelero, 2013) and the California Consumer Privacy Act (CCPA) (Pardau, 2018) enforce the *right to be forgotten*, requiring that user data and its influence on trained models be removed upon request. In practice, this requirement is difficult to satisfy, as user interactions are deeply entangled with model parameters (Zhao et al., 2024). Consequently, machine unlearning (Bourtoule et al., 2021; Nguyen et al., 2025) has emerged as a promising research direction for removing the influence of specific user data, referred to as the deletion set, from trained models.

To achieve effective machine unlearning, a method must address three core criteria (Chen et al., 2024). (i) Completeness: Ensuring that the influence of the deletion set is fully eliminated from the model. (ii) Utility: Preserving the model's performance on the remaining data. (iii) Efficiency: Enabling fast and computationally inexpensive unlearning compared to full retraining. Balancing these three objectives remains a central challenge in the design of practical unlearning systems.

**Recent Work and its limitations**. In recent years, there has been a few notable attempts devoted to unlearning in recommendation systems. Sharding-based methods, such as SISA (Bourtoule et al., 2021), RecEraser (Chen et al., 2022), and UltraRE (Li et al., 2023a), localize data influence by training multiple models, enabling unlearning through partial retraining. However, sharding weakens collaborative learning and incurs significant overhead under repeated deletion requests. Exact unlearning approaches (Xu et al., 2023; Zhang et al., 2023; Schelter et al., 2023) achieve retraining-equivalent updates via closed-form solutions, though they are restricted to specific model families and scale poorly with frequent deletions. More recently, influence-function-based methods such as SCIF (Chen et al., 2024) and IFRU (Zhang et al., 2024) have gained attention for approximate unlearning, leveraging second-order information to estimate and remove the influence of deletion set.

Despite these recent advancements, current recommendation unlearning methods still suffer from several notable limitations: **1.** Influence-based approaches require iterative computation of Hessian–vector products or other second-order statistics, making the process highly inefficient and limiting scalability for large models or frequent unlearning requests. **2.** While such methods often achieve good completeness by effectively removing the influence of the deleted data, they generally do not prioritize utility preser-

vation, leading to degraded recommendation performance on retained data after unlearning. **3.** Conversely, models like RRL (You et al., 2024) focus on maintaining utility by explicitly reversing the learning of deletion set but they fall short in achieving true completeness, as residual influences from the deleted interactions can remain in model parameters.

**Our contribution** To address the aforementioned limitations, we propose **Obliviate**, an efficient two-stage unlearning framework for recommendation systems that achieves high levels of completeness while maintaining good utility. As illustrated in Fig. 1, a model trained on the full dataset, $\theta_0$, and a model retrained after removing the deletion set, $\theta^{-S}$, can converge to different optima in the parameter space despite starting from the same initialization, $\theta_i$. The goal of unlearning is to efficiently transform $\theta_0$ into $\theta_{-S} \approx \theta_{-S}^*$ without full retraining by estimating the parameter change $\Delta\theta$ induced by the deletion set.

**Stage I: Low-Rank Unlearning Adapter (LUA)** In the first stage of **Obliviate**, We perform a Newton-style downdate (Liu et al., 2022; Bui et al., 2024) that reverses the immediate influence of the deletion set and downdate the model parameters back toward the local optimization state (denoted as noise point in Fig. 1). This stage performs an intermediate downdate that removes the dominant first-order and curvature-aware effects of the deletion set. To avoid the computational burden of inverting the full Hessian for downdate, we propose a positive-definite surrogate that enables efficient estimation of the update direction while preserving the key curvature information required for unlearning. Since applying downdate to the entire parameter space is prohibitively expensive, we initialize a *low-rank adapter* using the estimated downdate and restrict the correction to the most affected parameter subspaces. Moreover, to account for the collaborative nature of recommendation systems, we additionally propagate unlearning to neighboring users and items associated with the deletion set.

**Stage II: Locality-Aware Calibration (LAC)** While Stage I effectively reverses the dominant parameter drift induced by the deletion set and achieves high-level unlearning completeness, the aggressive downdate updates may introduce a utility gap in the recommendation quality. To address this issue, we introduce a lightweight calibration stage that refines only the low-rank adapters while keeping the backbone model parameters frozen. Specifically, LAC jointly optimizes two complementary objectives: (i) An unlearning objective explicitly suppresses the deleted interactions by driving their predicted scores toward negative values, ensuring that these samples no longer contribute to future optimization dynamics or collaborative preference formation. (ii) A utility-preserving distillation objective transfers predictive behavior from the original model ($\theta_0$) to

the calibrated unlearned model ($\theta_{-S}$) using a small subset of retained interactions. This distillation step restores ranking consistency and mitigates performance degradation introduced during downdate.

**Our contributions are summarized as follows:**

- We propose **Obliviate**, a novel unlearning method for recommender systems that achieves strong forgetting completeness while preserving recommendation utility.

- We introduce an efficient curvature proxy to approximate second-order information without explicitly computing the expensive Hessian matrix. Building on this, *LUA* performs a Newton-style downdate in a compact subspace, efficiently reversing the influence of deletion set while propagating unlearning to related users and items.

- We develop a lightweight *LAC* that refines the downdated parameters for better utility. LAC pushes deletion interactions toward negative preference signals while using distillation on retained data to prevent global utility drift.

- Extensive experiments across multiple benchmarks show that **Obliviate** achieves state-of-the-art unlearning performance even under large deletion ratios (as high as 20% of the data), delivering up to 9% relative utility gains while being up to 3× faster than strong retraining-based baselines.

## 2. Methodology

**Problem Formulation**

$$J(\theta) = \sum_{(u,i,y_{ui}) \in D} \ell\big(s(u,i;\theta), y_{ui}\big) + \Omega(\theta), \quad (1)$$

where $J(\theta)$ denotes the empirical training objective of a differentiable recommender model parameterized by $\theta \in \mathbb{R}^p$, $s(u,i;\theta)$ is the predicted score for user-item pair $(u,i)$ from dataset $D$, $\ell(\cdot)$ is the training loss, and $\Omega(\theta)$ is a regularization term. The above objective gives $\theta_0 = \arg\min_\theta J(\theta)$.

Given a deletion set $S \subset D$, the retraining objective after removing $S$ is:

$$J_{-S}(\theta) = \sum_{(u,i,y_{ui}) \in D \setminus S} \ell\big(s(u,i;\theta), y_{ui}\big) + \Omega(\theta), \quad (2)$$

which minimized as $\theta^\star_{-S} = \arg\min_\theta J_{-S}(\theta)$. on retained set $D/S$.

The objective of machine unlearning is to obtain an updated model as:

$$\theta_{-S} = \theta_0 + \Delta\theta, \quad (3)$$

such that $\theta_{-S} \approx \theta^\star_{-S}$ without performing full retraining. Here, $\Delta\theta$ represents the parameter shift induced by removing the deletion set $S$. Rather than optimizing $J_{-S}$ from scratch, we estimate $\Delta\theta$ by characterizing the local change in the optimum $\theta_0$ after removing the contribution of the deletion set $S$.

**Proposed Method: Obliviate**

We propose a two-stage, architecture-agnostic unlearning framework consisting of (1) A *Low-Rank Unlearning Adapter* (LUA) Stage and (2) *Locality-Aware Calibration* (LAC) Stage as shown in Fig. 2. Our objective is to approximate the solution obtained by retraining on the retained dataset $D \setminus S$ while avoiding full retraining.

Unlearning can be interpreted as a *counterfactual empirical risk minimization* problem (Koh & Liang, 2017). Let $J(\theta)$ be the empirical objective over dataset $D$ and $J_{-S}(\theta) = J(\theta) - J_S(\theta)$ be the objective with subset $S$ removed. The transition from $\theta_0 = \arg\min J(\theta)$ to $\theta^\star_{-S} = \arg\min J_{-S}(\theta)$ can be viewed as a local sensitivity analysis problem, where the optimizer shift is determined by the deletion gradient $\nabla J_S(\theta_0)$ and the local curvature $\nabla^2 J(\theta_0)$. Our framework operationalizes this sensitivity estimate in a structured and computationally efficient manner.

### 2.1. Low-Rank Unlearning Adapter (LUA)

#### 2.1.1. DELETING GRADIENT

We define the aggregate deletion gradient associated with the deletion set $S$ as

$$g_S = \sum_{(u,i,y_{ui}) \in S} \nabla_\theta \ell(s(u,i;\theta_0), y_{ui}) = \nabla J_S(\theta_0), \quad (4)$$

which measures the first-order influence of the deletion set on the learned parameters through its contribution to the stationarity (optimum) condition at $\theta_0$.

Since $\nabla J(\theta_0) = 0$ at the optimum and $J(\theta) = J_{-S}(\theta) + J_S(\theta)$, therefore, $\nabla_\theta J(\theta) = \nabla_\theta J_{-S}(\theta) + \nabla_\theta J_S(\theta)$ for optimum $\nabla_\theta J_{-S}(\theta_0) + \nabla_\theta J_S(\theta_0) = 0 => \nabla_\theta J_{-S}(\theta_0) + gs = 0 => J_{-S}(\theta_0) = -gs$ removing $S$ induces a shift in the stationarity condition by $-g_S$. Thus, unlearning reduces to compensating for the resulting shift in parameter space.

#### 2.1.2. SECOND-ORDER DOWNDATE

To estimate the parameter shift required to transform $\theta_0$ into $\theta_{-S}$, a second-order approximation of the objective around the full-data optimum is required. This yields a curvature-aware downdate that captures the influence of the deletion set $S$ (Bonnans & Shapiro, 2000; Bui et al., 2024).

**Theorem 2.1.** *[Second-Order Unlearning Approximation] Assume $J(\theta)$ is twice differentiable and locally strongly*

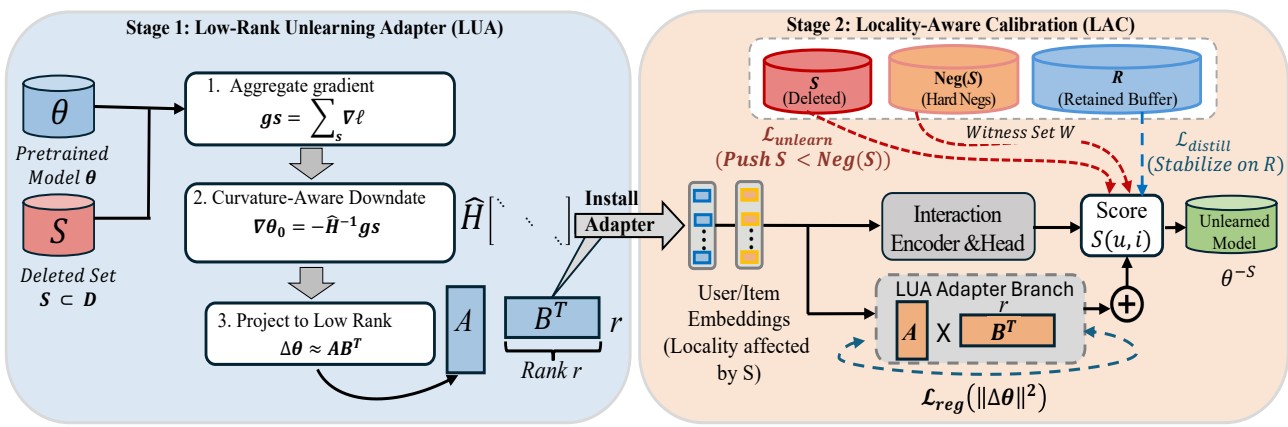

*Figure 2.* **Overview of Obliviate. LUA** performs a curvature-aware low-rank downdate using deletion set $S$ to localize forgetting. **LAC** then refines the model with deleted samples, hard negatives, and retained data, combining unlearning and distillation losses to produce the final model $\theta^{-S}$ with preserved utility.

*convex in a neighborhood of $\theta_0$. Let*

$$H = \nabla_\theta^2 J(\theta_0)$$

*be the Hessian of the full-data objective. Then, under a second-order Taylor approximation,*

$$\theta_{-S}^\star \approx \theta_0 - H^{-1} g_S. \tag{5}$$

*Proof.* Since $\theta_0$ minimizes $J(\theta)$, we have $\nabla J(\theta_0) = 0$. Decompose $J(\theta) = J_{-S}(\theta) + J_S(\theta)$. Then

$$\nabla J_{-S}(\theta_0) = \nabla J(\theta_0) - \nabla J_S(\theta_0) = -g_S.$$

Applying second-order Taylor expansion of $J_{-S}$ around $\theta_0$:

$$J_{-S}(\theta_0 + \Delta) \approx J_{-S}(\theta_0) + \nabla J_{-S}(\theta_0)^\top \Delta + \frac{1}{2}\Delta^\top H\Delta,$$

where we approximate $\nabla^2 J_{-S}(\theta_0) \approx H$ since $S$ is small. Minimizing this quadratic approximation yields

$$-g_S + H\Delta = 0 \quad \Rightarrow \quad \Delta = H^{-1}g_S, \tag{6}$$

hence $\theta_{-S}^\star \approx \theta_0 - H^{-1}g_S.$ □

A quantitative error bound for this approximation is provided in Appendix K.3.

**Local Convexity Assumption:** The above derivation requires strong convexity only in a local neighborhood around the trained model $\theta_0$, rather than globally across the entire parameter space. Although recommender models such as LightGCN are non-convex in general, the loss landscape near a well-trained solution often exhibits approximately quadratic behavior. Under this local regime, the approximation $\theta_{-S}^\star \approx \theta_0 - H^{-1}g_S$ provides a useful estimate of the retraining-induced parameter shift. Similar local sensitivity

assumptions are commonly adopted in influence-function and optimization-based analyses of deep models (Jin et al., 2017; Milne, 2019; Pun & So, 2021). Additional discussion is provided in Appendix I.

### 2.1.3. PRACTICAL CURVATURE APPROXIMATION

As shown in Theorem 2.1, estimating the retraining-induced parameter shift requires the curvature-corrected update $H^{-1}g_S$. Since computing or inverting the full Hessian is intractable for large recommender models. We therefore propose a diagonal preconditioner derived from Adam second-moment statistics, which provides a computationally efficient and positive-definite surrogate for local curvature:

$$(\widehat{H}^{-1}g_S)_j \approx \frac{(g_S)_j}{\sqrt{\hat{v}_j} + \varepsilon}. \tag{7}$$

Let $\widehat{H} = \text{diag}(\sqrt{\hat{v}} + \varepsilon)$. Then

$$\Delta\theta_0 = \arg\min_\Delta \ g_S^\top \Delta + \frac{1}{2}\Delta^\top \widehat{H}\Delta, \tag{8}$$

This corresponds to steepest descent under the weighted norm $\|\Delta\|_{\widehat{H}}^2 = \Delta^\top \widehat{H}\Delta$. Since $\hat{v}_j \approx \mathbb{E}[g_j(\theta)^2]$, Adam's second-moment statistics act as a data-adaptive diagonal curvature proxy. Reusing these statistics provides a stable, curvature-aware preconditioner without extra computation (see Appendix H.2 - H.4).

### 2.1.4. LOW-RANK ADAPTER PROJECTION

Directly applying $\Delta\theta_0$ to all parameters is inefficient and unnecessary. We instead solve

$$\min_{\text{rank}(\Delta W) \leq r} \|\Delta W - \Delta W_0\|_F^2, \tag{9}$$

whose solution is given by truncated SVD (Hu et al., 2022; Hansen, 1987; Maheri et al., 2025).

**Algorithm 1 Obliviate**: Two-Stage Unlearning with Low-Rank Adapters

---

**Require:** Trained model $\theta_0$; deletion set $S$; rank $r$; curvature proxy $\widehat{H}^{-1}$; negatives $q(i \mid u)$; retained buffer $R$; LAC steps $T$; lr $\eta$.
**Ensure:** Unlearned model $\theta_{-S} = (\theta_0, \phi^\star)$.
1: **LUA:** Compute $g_S$ (Eq. 4) and downdate $\Delta\theta_0$ (Eq. 6 with proxy Eq. 8).
2: **for** each selected block $W$ **do**
3:     Extract $\Delta W_0$ from $\Delta\theta_0$; factorize $\Delta W_0 \approx AB^\top$ (Eq. 10).
4:     Initialize adapter params $\phi_W \leftarrow (A, B)$; freeze $W$ and set $W' = W + AB^\top$ (Eq. 11).
5: **end for**
6: $\phi \leftarrow \{\phi_W\}_{W \in \mathcal{B}}$.
7: **LAC:** Construct witness set $W$ (Eq. 12) by sampling negatives for $S$.
8: **for** $t = 1$ to $T$ **do**
9:     Sample mini-batches $B_S \subset \mathcal{T}_S$, $B_R \subset R$.
10:     Compute $\mathcal{L}(\phi)$ (Eq. 13) using Eqs. 14, 15 .
11:     Update adapters: $\phi \leftarrow \phi - \eta\nabla_\phi\mathcal{L}(\phi)$.
12: **end for**
13: **return** $\theta_{-S} = (\theta_0, \phi^\star)$.

---

**Proposition 2.2** (Optimal Low-Rank Projection). *Let $\Delta W_0 = U\Sigma V^\top$ be the SVD. The best rank-$r$ approximation in Frobenius norm is $\Delta W^\star = U_r\Sigma_r V_r^\top$.*

We parameterize this update as

$$\Delta W = AB^\top, \quad A = U_r\Sigma_r^{1/2}, \quad B = V_r\Sigma_r^{1/2}. \quad (10)$$

The adapted block becomes

$$W' = W + AB^\top, \quad (11)$$

while the base model remains frozen. (see Appendix H.5).

In recommender systems, deletion effects are largely low-dimensional, mainly impacting a small subset of user and item embeddings. Thus, the retraining-induced parameter shift is approximately low-rank (Hanada et al., 2023). Projecting the Newton update into a low-rank subspace captures the key forgetting directions while reducing cost, and representing it as an adapter makes unlearning modular and reversible. Additional detail is provided in Appendix H.6.

Theoretical analysis in Appendix K.12 establishes an end-to-end approximation bound for LUA, decomposing the error into (i) a second-order sensitivity remainder, (ii) curvature approximation error, and (iii) low-rank projection error.

**2.2. Locality-Aware Calibration (LAC)**

While LUA provides an efficient approximation to the retraining solution via downdate, approximation and projec-

tion errors may introduce a utility gap. LAC addresses this through a lightweight refinement of the low-rank adapters, improving utility while preserving unlearning effectiveness.

Let $\phi$ denote the adapter parameters and let $\theta(\phi) = \theta_0 + \Delta\theta(\phi)$. LUA provides an initialization $\phi_0$ corresponding to the projected second-order downdate. Starting from this initialization, LAC optimizes the adapter parameters to further improve the approximation of $\theta(\phi^\star) \approx \theta^\star_{-S}$ while maintaining utility on retained data.

Since the backbone model $\theta_0$ remains frozen, optimization is restricted to the low-dimensional adapter subspace $\mathcal{A} = \{\Delta\theta(\phi)\}$, ensuring that $\theta(\phi) - \theta_0 \in \mathcal{A}$. This locality constraint preserves the modular nature of unlearning and limits parameter drift outside the subspace identified by LUA.

While the ideal objective would optimize over the entire retained dataset, doing so would substantially increase the computational cost of unlearning. We therefore construct a compact *witness set* that captures the key signals required for forgetting and utility preservation, and use it as a surrogate objective during calibration.

2.2.1. WITNESS SET

We define the witness set as:

$$W = S \cup \text{Neg}(S) \cup R, \quad (12)$$

where $S$ denotes the deletion set, $\text{Neg}(S)$ denotes hard negative interactions associated with $S$, and $R$ is a small buffer of retained interactions.

Each component serves a distinct role: the deletion set enforces forgetting, the hard negatives preserve local ranking structure, and the retained buffer maintains recommendation utility on unaffected data. Together, these samples provide a compact surrogate for the full retraining objective, enabling efficient calibration without revisiting the entire dataset. Under standard smoothness assumptions, controlling the calibration objective on this representative subset limits deviation on the retained data distribution. Additional discussion is provided in Appendix J.1.

2.2.2. CALIBRATION OBJECTIVE

LAC optimizes the following loss $\mathcal{L}_{net}$ which is the joint loss of unlearning and distillation:

$$\mathcal{L}_{net} = \min_\phi \Big[\lambda_{\text{unlearn}}\mathcal{L}_{\text{unlearn}} + \lambda_{\text{distill}}\mathcal{L}_{\text{distill}} \\ + \lambda_{\text{reg}}\|\Delta\theta(\phi)\|_F^2\Big]. \quad (13)$$

**Unlearning Loss.** We define the unlearning objective as $\mathcal{L}_{\text{unlearn}} = \ell_{\text{BPR}}$ where the BPR loss is given by

$$\ell_{\text{BPR}}(u, i^+, i^-) = -\log \sigma\big(s(u, i^-; \theta(\phi)) - s(u, i^+; \theta(\phi))\big). \tag{14}$$

where $i^+$ is a deleted item and $i^-$ is a sampled negative item. Minimizing $\ell_{\text{BPR}}$ encourages the deleted item to be ranked below the negative item, thereby removing its influence from the recommendation model.

**Distillation Loss.** To preserve recommendation quality on retained interactions, we match the predictions of the unlearned model to those of the original model on a small retained buffer $R$:

$$\mathcal{L}_{\text{distill}} = \frac{1}{|R|} \sum_{(u,i) \in R} \|s(u, i; \theta(\phi)) - s(u, i; \theta_0)\|_2^2. \tag{15}$$

The LAC objective balances three complementary goals: (i) forgetting deleted interactions, (ii) preserving utility on retained data, and (iii) restricting updates to remain local to the adapter subspace. This yields a stable refinement of the LUA solution without requiring full retraining.

LAC contributes optimization descent (Theorem K.14) and a locality/drift certificate (Corollary K.15). Finally, Proposition K.17 and Proposition K.18 translate parameter locality into score-level utility and forgetting guarantees.

### 2.3. Interpretation as Trust-Region Optimization

**Proposition 2.3.** *The LAC objective is equivalent to a trust-region refinement around the LUA solution, where the regularization term bounds the magnitude of the correction in parameter space.*

Thus, LAC performs a localized second-order correction that improves utility while preserving the unlearning guarantees of LUA. The trust-region view explains why LAC remains stable. Rather than freely re-optimizing, LAC restricts updates to remain close to the LUA solution in parameter space. This ensures that calibration improves the approximation to $\theta^\star_{-S}$ while preserving the modular and localized nature of unlearning. Appendix J can be referred alongside.

## 3. Experiments

### 3.1. Research Questions

**RQ1: Utility Preservation.** How closely does the recommendation performance of the unlearned model match that of a model retrained from scratch after removing the deletion set?

**RQ2: Unlearning Completeness.** To what extent does the

proposed method remove the influence of the deletion set $S$ from the unlearned model?

**RQ3: Computational Efficiency.** How efficiently does the proposed method perform unlearning in terms of computation time and resource consumption compared to full retraining and existing unlearning baselines?

**RQ4: Component Contribution.** How does each component contribute to the effectiveness of the proposed unlearning framework?

### 3.2. Simulation of Unlearning Scenario

To reliably assess whether unlearning is successfully achieved, the design of the deletion simulation is critical. We define a set of user–item interactions to be removed, referred to as the *deletion set $S$*. A naive approach would randomly sample existing interactions as deletion requests; however, removing genuine data perturbs the collaborative structure of the dataset, confounding the effects of unlearning with degradation due to the loss of true preference signals.

To avoid this confounding effect, we adopt a synthetic deletion protocol. We randomly select a subset of users and inject additional interactions with items they never interacted with and for which predicted preference scores are low. These injected interactions simulate spurious feedback and are designated as deletion data. This design enables us to isolate the impact of unlearning while preserving the original collaborative structure of the dataset. A justification of this protocol is provided in Appendix F.

**How better is Obliviate in imitating real world unlearning setup**: In all experiments, we simulate unlearning requests from 20% of users for all datasets and baselines, treating their added interactions as the deletion set. This relatively high deletion ratio provides a more stringent and realistic evaluation of unlearning effectiveness compared to prior work that typically considers only 1–5% deletion. Such small fractions are insufficient to stress-test a method's ability to remove large-scale influence, whereas our setting better reflects real-world scenarios where substantial portions of user data may need to be forgotten.

### 3.3. Training, Data, and Implementation Details

For all experiments, we use two widely adopted recommendation backbones: MF-BPR and LightGCN. To comprehensively evaluate performance, we compare our method against strong baselines, including sharding-based approaches (SISA and RecEraser) as well as recent state-of-the-art unlearning methods (IFRU and RRL). Detailed training configurations and implementation specifics are provided in Appendix A- D.

**RQ1: Utility Preservation.** **Obliviate** consistently

*Table 1.* **Recommendation performance across ML-1M, Amazon, and Yelp.** We report Recall (R@K) and NDCG (N@K) before unlearning (Original), after exact retraining on retained data (Retrained), and after applying unlearning methods (Unlearned). Δ% denotes the relative improvement of the Unlearned model over the Original model. Higher is better.

| | Methods | Phase | MF-BPR | | | | | | LightGCN | | | | | |
|---|---|---|---|---|---|---|---|---|---|---|---|---|---|---|
| | | | R@10 | R@20 | R@50 | N@10 | N@20 | N@50 | R@10 | R@20 | R@50 | N@10 | N@20 | N@50 |
| **ML-1M** | SISA | Retrained | – | – | – | – | – | – | – | – | – | – | – | – |
| | | Original | 0.1139 | 0.1831 | 0.3152 | 0.2843 | 0.2741 | 0.2948 | 0.1016 | 0.1629 | 0.2868 | 0.2910 | 0.2743 | 0.2859 |
| | | Unlearned | 0.1209 | 0.1928 | 0.3313 | 0.2987 | 0.2876 | 0.3099 | 0.1067 | 0.1701 | 0.3006 | 0.3106 | 0.2912 | 0.3020 |
| | | Δ% | +6.15 | +5.30 | +5.11 | +5.07 | +4.93 | +5.12 | +5.02 | +4.42 | +4.81 | +6.74 | +6.16 | +5.63 |
| | RecEraser | Retrained | – | – | – | – | – | – | – | – | – | – | – | – |
| | | Original | 0.0809 | 0.1328 | 0.2421 | 0.2399 | 0.2266 | 0.2387 | 0.0743 | 0.1249 | 0.2278 | 0.2209 | 0.2117 | 0.2233 |
| | | Unlearned | 0.0829 | 0.1371 | 0.2463 | 0.2457 | 0.2337 | 0.2446 | 0.0782 | 0.1316 | 0.2374 | 0.2383 | 0.2268 | 0.2365 |
| | | Δ% | +2.47 | +3.24 | +1.73 | +2.42 | +3.13 | +2.47 | +5.25 | +5.36 | +4.21 | +7.88 | +7.13 | +5.91 |
| | IFRU | Retrained | 0.1179 | 0.1888 | 0.3299 | 0.3320 | 0.3144 | 0.3280 | 0.1363 | 0.2267 | 0.3752 | 0.3450 | 0.3422 | 0.3520 |
| | | Original | 0.1123 | 0.1795 | 0.3098 | 0.3109 | 0.2946 | 0.3072 | 0.1223 | 0.2071 | 0.3421 | 0.3136 | 0.3202 | 0.3321 |
| | | Unlearned | 0.1127 | 0.1803 | 0.3112 | 0.3126 | 0.2962 | 0.3088 | 0.1237 | 0.2098 | 0.3458 | 0.3172 | 0.3238 | 0.3355 |
| | | Δ% | +0.36 | +0.45 | +0.45 | +0.55 | +0.54 | +0.52 | +1.14 | +1.30 | +1.08 | +1.15 | +1.12 | +1.02 |
| | RRL | Retrained | 0.1265 | 0.2023 | 0.3465 | 0.3404 | 0.3241 | 0.3399 | 0.1556 | 0.2429 | 0.4043 | 0.3878 | 0.3710 | 0.3909 |
| | | Original | 0.1190 | 0.1890 | 0.3262 | 0.3204 | 0.3043 | 0.3191 | 0.1444 | 0.2306 | 0.3862 | 0.3584 | 0.3472 | 0.3684 |
| | | Unlearned | 0.1196 | 0.1897 | 0.3264 | 0.3239 | 0.3073 | 0.3208 | 0.1455 | 0.2321 | 0.3885 | 0.3607 | 0.3487 | 0.3699 |
| | | Δ% | +0.50 | +0.37 | +0.06 | +1.09 | +0.99 | +0.53 | +0.76 | +0.65 | +0.60 | +0.64 | +0.43 | +0.41 |
| | **Obliviate** | Retrained | 0.1265 | 0.2023 | 0.3465 | 0.3404 | 0.3241 | 0.3399 | 0.1556 | 0.2429 | 0.4043 | 0.3878 | 0.3710 | 0.3909 |
| | | Original | 0.1190 | 0.1890 | 0.3262 | 0.3204 | 0.3043 | 0.3191 | 0.1444 | 0.2306 | 0.3862 | 0.3584 | 0.3472 | 0.3684 |
| | | Unlearned | 0.1210 | 0.1932 | 0.3326 | 0.3279 | 0.3117 | 0.3263 | 0.1460 | 0.2332 | 0.3903 | 0.3623 | 0.3506 | 0.3713 |
| | | Δ% | **+1.68** | **+2.22** | **+1.96** | **+2.34** | **+2.43** | **+2.26** | **+1.11** | **+1.13** | **+1.06** | **+1.09** | **+0.98** | **+0.79** |
| **Amazon** | SISA | Retrained | – | – | – | – | – | – | – | – | – | – | – | – |
| | | Original | 0.0070 | 0.0123 | 0.0206 | 0.0054 | 0.0074 | 0.0099 | 0.0081 | 0.0134 | 0.0255 | 0.0061 | 0.0080 | 0.0116 |
| | | Unlearned | 0.0068 | 0.0115 | 0.0199 | 0.0054 | 0.0071 | 0.0097 | 0.0102 | 0.0174 | 0.0320 | 0.0078 | 0.0104 | 0.0147 |
| | | Δ% | -2.86 | -6.50 | -3.40 | 0.00 | -4.05 | -2.02 | +25.93 | +29.85 | +25.49 | +27.87 | +30.00 | +26.72 |
| | RecEraser | Retrained | – | – | – | – | – | – | – | – | – | – | – | – |
| | | Original | 0.0071 | 0.0114 | 0.0204 | 0.0051 | 0.0067 | 0.0094 | 0.0051 | 0.0088 | 0.0178 | 0.0035 | 0.0049 | 0.0074 |
| | | Unlearned | 0.0076 | 0.0122 | 0.0211 | 0.0057 | 0.0074 | 0.0101 | 0.0081 | 0.0129 | 0.0245 | 0.0059 | 0.0077 | 0.0110 |
| | | Δ% | +7.04 | +7.02 | +3.43 | +11.76 | +10.45 | +7.45 | +58.82 | +46.59 | +37.64 | +68.57 | +57.14 | +48.65 |
| | IFRU | Retrained | 0.0125 | 0.0214 | 0.0413 | 0.0095 | 0.0128 | 0.0185 | 0.0131 | 0.0234 | 0.0417 | 0.0101 | 0.0139 | 0.0191 |
| | | Original | 0.0091 | 0.0169 | 0.0330 | 0.0063 | 0.0092 | 0.0140 | 0.0107 | 0.0195 | 0.0342 | 0.0077 | 0.0101 | 0.0151 |
| | | Unlearned | 0.0094 | 0.0176 | 0.0335 | 0.0065 | 0.0096 | 0.0143 | 0.0109 | 0.0198 | 0.0345 | 0.0079 | 0.0103 | 0.0153 |
| | | Δ% | +3.30 | +4.14 | +1.51 | +3.17 | +4.34 | +2.14 | +1.87 | +1.54 | +0.88 | +2.60 | +1.98 | +1.32 |
| | RRL | Retrained | 0.0136 | 0.0226 | 0.0464 | 0.0103 | 0.0136 | 0.0204 | 0.0142 | 0.0242 | 0.0460 | 0.0103 | 0.0140 | 0.0202 |
| | | Original | 0.0122 | 0.0218 | 0.0419 | 0.0090 | 0.0125 | 0.0184 | 0.0123 | 0.0215 | 0.0397 | 0.0086 | 0.0119 | 0.0174 |
| | | Unlearned | 0.0127 | 0.0226 | 0.0421 | 0.0092 | 0.0128 | 0.0185 | 0.0125 | 0.0218 | 0.0399 | 0.0087 | 0.0121 | 0.0174 |
| | | Δ% | +4.10 | +3.67 | +0.48 | +2.22 | +2.40 | +0.54 | +1.63 | +1.40 | +0.50 | +1.16 | +1.68 | 0.00 |
| | **Obliviate** | Retrained | 0.0136 | 0.0226 | 0.0464 | 0.0103 | 0.0136 | 0.0204 | 0.0142 | 0.0242 | 0.0460 | 0.0103 | 0.0140 | 0.0202 |
| | | Original | 0.0122 | 0.0218 | 0.0419 | 0.0090 | 0.0125 | 0.0184 | 0.0123 | 0.0215 | 0.0397 | 0.0086 | 0.0119 | 0.0174 |
| | | Unlearned | 0.0134 | 0.0232 | 0.0431 | 0.0097 | 0.0133 | 0.0190 | 0.0127 | 0.0221 | 0.0403 | 0.0089 | 0.0123 | 0.0176 |
| | | Δ% | **+9.84** | **+6.42** | **+2.86** | **+7.78** | **+6.40** | **+3.26** | **+3.25** | **+2.79** | **+1.51** | **+3.49** | **+3.36** | **+1.15** |
| **Yelp** | SISA | Retrained | – | – | – | – | – | – | – | – | – | – | – | – |
| | | Original | 0.0207 | 0.0356 | 0.0706 | 0.0242 | 0.0294 | 0.0425 | 0.0198 | 0.0349 | 0.0696 | 0.0225 | 0.0281 | 0.0410 |
| | | Unlearned | 0.0219 | 0.0381 | 0.0760 | 0.0259 | 0.0317 | 0.0457 | 0.0235 | 0.0406 | 0.0787 | 0.0270 | 0.0332 | 0.0473 |
| | | Δ% | +5.80 | +7.02 | +7.65 | +7.02 | +7.82 | +7.53 | +18.69 | +16.33 | +13.07 | +20.00 | +18.15 | +15.37 |
| | RecEraser | Retrained | – | – | – | – | – | – | – | – | – | – | – | – |
| | | Original | 0.0116 | 0.0210 | 0.0447 | 0.0135 | 0.0170 | 0.0259 | 0.0098 | 0.0198 | 0.0456 | 0.0110 | 0.0151 | 0.0249 |
| | | Unlearned | 0.0173 | 0.0297 | 0.0587 | 0.0202 | 0.0246 | 0.0354 | 0.0175 | 0.0304 | 0.0610 | 0.0208 | 0.0254 | 0.0367 |
| | | Δ% | +49.14 | +41.43 | +31.31 | +49.63 | +44.71 | +36.68 | +78.57 | +53.54 | +33.77 | +89.09 | +68.21 | +47.39 |
| | IFRU | Retrained | 0.0178 | 0.0310 | 0.0611 | 0.0208 | 0.0255 | 0.0367 | 0.0191 | 0.0371 | 0.0971 | 0.0235 | 0.0296 | 0.0612 |
| | | Original | 0.0126 | 0.0231 | 0.0442 | 0.0150 | 0.0189 | 0.0267 | 0.0251 | 0.0435 | 0.0876 | 0.0290 | 0.0357 | 0.0521 |
| | | Unlearned | 0.0127 | 0.0233 | 0.0445 | 0.0152 | 0.0191 | 0.0269 | 0.0257 | 0.0444 | 0.0889 | 0.0295 | 0.0363 | 0.0528 |
| | | Δ% | +0.79 | +0.87 | +0.68 | +1.33 | +1.06 | +0.75 | +2.39 | +2.07 | +1.48 | +1.72 | +1.68 | +1.34 |
| | RRL | Retrained | 0.0212 | 0.0361 | 0.0708 | 0.0251 | 0.0303 | 0.0432 | 0.0323 | 0.0559 | 0.1103 | 0.0364 | 0.0452 | 0.0656 |
| | | Original | 0.0191 | 0.0319 | 0.0620 | 0.0224 | 0.0268 | 0.0379 | 0.0288 | 0.0514 | 0.1043 | 0.0328 | 0.0413 | 0.0611 |
| | | Unlearned | 0.0190 | 0.0322 | 0.0631 | 0.0224 | 0.0269 | 0.0383 | 0.0293 | 0.0520 | 0.1052 | 0.0334 | 0.0419 | 0.0619 |
| | | Δ% | -0.52 | +0.94 | +1.77 | 0.00 | +0.37 | +1.06 | +1.74 | +1.17 | +0.86 | +1.83 | +1.45 | +1.31 |
| | **Obliviate** | Retrained | 0.0212 | 0.0361 | 0.0708 | 0.0251 | 0.0303 | 0.0432 | 0.0323 | 0.0559 | 0.1103 | 0.0364 | 0.0452 | 0.0656 |
| | | Original | 0.0191 | 0.0319 | 0.0620 | 0.0224 | 0.0268 | 0.0379 | 0.0288 | 0.0514 | 0.1043 | 0.0328 | 0.0413 | 0.0611 |
| | | Unlearned | 0.0197 | 0.0331 | 0.0646 | 0.0230 | 0.0276 | 0.0392 | 0.0297 | 0.0530 | 0.1068 | 0.0337 | 0.0423 | 0.0629 |
| | | Δ% | **+3.14** | **+3.76** | **+4.19** | **+2.68** | **+2.99** | **+3.43** | **+3.13** | **+3.11** | **+2.40** | **+2.74** | **+2.42** | **+2.95** |

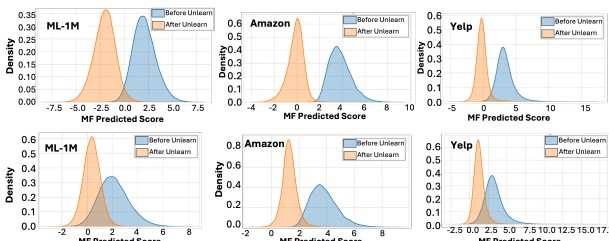

*Figure 3.* Completeness: The top row shows a hard unlearning setting with increased calibration and the bottom row shows when the utility is maintained.

improves recommendation performance after unlearning across all datasets and backbones (Table 1). On ML-1M, it delivers stable gains over the original model, while IFRU and RRL show only marginal changes. The improvement is more pronounced on sparse datasets (Amazon, Yelp), where **Obliviate** achieves the largest and most consistent gains, demonstrating strong preservation of collaborative signals. This stems from our two-stage design: LUA confines updates to a low-rank subspace to avoid disruptive global shifts, while LAC restores ranking fidelity via lightweight distillation on retained data. Together, they maintain utility close to retraining with significantly lower cost.

**Why do baseline models start from different performance levels?** Baselines use different training and unlearning strategies, which lead to varying original model quality. To ensure fair evaluation, we report the *relative percentage change* from each model's own original performance rather than comparing absolute scores.

**Why do SISA and RecEraser show larger gains?** They rely on data sharding and independent sub-model training. Since unlearning requests often span multiple shards, many sub-models must be retrained, effectively approximating full retraining. This yields stronger utility recovery but at substantially higher computational cost, limiting practicality in large-scale or frequent unlearning settings.

**RQ2: Completeness. Obliviate** not only preserves recommendation utility but also effectively removes the direct influence of the deletion data. As shown in Fig. 3, the predicted score distribution of deletion-set interactions shifts markedly after unlearning (blue → orange), indicating reduced model preference for removed data. The top row illustrates a *hard forgetting* setting with increased calibration steps in LAC, resulting in stronger completeness. The bottom row corresponds to the configuration used for utility evaluation, which achieves a better utility–forgetting balance while still demonstrating clear influence removal. Appendix G reports the Demotion Rate before and after unlearning for both recommendation backbones across all datasets.

*Table 2.* **Training and unlearning time comparison (seconds).** Retraining denotes training from scratch on the retained dataset. Unlearning denotes the time required to remove the influence of deleted data from a trained model. Lower is better.

| Setting | Phase | SISA | RecEraser | IFRU | RRL | Obliviate |
|---|---|---|---|---|---|---|
| **ML-1M** | | | | | | |
| MF-BPR | Retraining | 504.00 | 483.50 | 975.00 | 533.50 | 533.50 |
| | Unlearning | 533.88 | 508.80 | 33.75 | 18.23 | 19.95 |
| LightGCN | Retraining | 1097.26 | 1121.11 | 1672.12 | 1181.94 | 1181.94 |
| | Unlearning | 1145.70 | 1181.37 | 42.87 | 21.72 | 24.32 |
| **Amazon** | | | | | | |
| MF-BPR | Retraining | 55.76 | 58.80 | 142.70 | 88.64 | 88.64 |
| | Unlearning | 58.87 | 68.80 | 23.76 | 12.25 | 13.55 |
| LightGCN | Retraining | 117.00 | 120.31 | 242.00 | 135.19 | 135.19 |
| | Unlearning | 122.98 | 122.63 | 33.21 | 15.31 | 17.07 |
| **Yelp** | | | | | | |
| MF-BPR | Retraining | 842.10 | 965.00 | 1575.00 | 896.00 | 896.00 |
| | Unlearning | 851.30 | 974.00 | 120.00 | 39.11 | 41.39 |
| LightGCN | Retraining | 1893.10 | 1922.00 | 2732.37 | 1984.45 | 1984.45 |
| | Unlearning | 1922.82 | 1926.91 | 157.00 | 54.61 | 57.23 |

**RQ3: Computational Efficiency**
**Obliviate** is substantially more efficient than both sharding-based and retraining-based unlearning approaches, as shown in Table 2. Across all datasets and backbones, SISA and RecEraser exhibit runtimes comparable to full retraining (often exceeding 500s on ML-1M and 1900s on Yelp with LightGCN), since all affected shards must be retrained and, in the worst case, a large portion of the data may need to be revisited. Influence-based methods such as IFRU reduce this cost but still incur noticeable overhead from repeated second-order computations.

In contrast, **Obliviate** consistently performs unlearning in seconds rather than minutes. For example, it requires only 19.9s versus 533s for retraining on ML-1M MF-BPR, 24.3s versus 1182s on ML-1M LightGCN, and 57.2s versus 1984s on Yelp LightGCN. This corresponds to more than an order-of-magnitude speedup over full retraining and sharding-based methods. Moreover, its runtime remains comparable to RRL and substantially lower than IFRU, while achieving stronger unlearning completeness and competitive or superior utility.

**Why is the unlearning time similar to retraining for SISA and RecEraser?** SISA and RecEraser rely on retraining affected shards after deletion. While efficient when only a few shards are involved, recommender system deletion requests often span interactions distributed across many shards. Consequently, a large fraction of the shards may need to be retrained, causing the computational cost to approach that of full retraining. Additional overhead from identifying affected shards further increases runtime, making practical unlearning time comparable to retraining from scratch.

*Table 3.* **Component ablation results.** Performance after unlearning when removing key components of **Obliviate**.

| Dataset | Variant | MF-BPR | | LightGCN | |
|---|---|---|---|---|---|
| | | R@20 | N@20 | R@20 | N@20 |
| ML-1M | No-LUA | 0.1921 | 0.3089 | 0.2321 | 0.3491 |
| | No-LAC | 0.1907 | 0.3062 | 0.2314 | 0.3480 |
| | w/o $\mathcal{L}_{\text{distill}}$ | 0.1922 | 0.3085 | 0.2319 | 0.3487 |
| Amazon | No-LUA | 0.0227 | 0.0130 | 0.0219 | 0.0121 |
| | No-LAC | 0.0221 | 0.0127 | 0.0217 | 0.0119 |
| | w/o $\mathcal{L}_{\text{distill}}$ | 0.0228 | 0.0131 | 0.0219 | 0.0122 |
| Yelp | No-LUA | 0.0327 | 0.0273 | 0.0524 | 0.0420 |
| | No-LAC | 0.0323 | 0.0270 | 0.0518 | 0.0416 |
| | w/o $\mathcal{L}_{\text{distill}}$ | 0.0326 | 0.0273 | 0.0523 | 0.0419 |

**RQ4: Component Contribution.**
In Table 3, we analyze the contribution of each component by removing key modules and evaluating performance on all datasets across both MF-BPR and LightGCN.

**No-LUA.** Removing the Low-Rank Unlearning Adapter forces the model to rely on direct parameter updates without structured low-rank constraints. This leads to less controlled unlearning updates that disturb collaborative representations. As shown in the table, performance consistently drops across all datasets and backbones, indicating its importance in unlearning.

**No-LAC.** When LAC stage is removed, the model performs only the initial downdate step. Although forgetting is still applied, the absence of calibration causes noticeable degradation in recommendation quality, especially in sparse datasets (Amazon and Yelp). This confirms that LAC is essential for restoring ranking. **w/o** $\mathcal{L}_{\text{distill}}$**.** Removing the distillation loss within LAC weakens the model's ability to align predictions on retained data with the original model. While performance remains better than removing LAC entirely, consistent declines across all settings show that knowledge distillation is a key driver of utility preservation, helping the model recover collaborative signals that might otherwise be weakened during unlearning.

Overall, these results demonstrate that each component contributes to the final performance, with *LAC playing a particularly important role in guiding the parameters toward the global minimum corresponding to retraining on retained data.* This calibration step ensures that Obliviate achieves strong utility preservation while maintaining effective unlearning.

## 4. Conclusion

We proposed **Obliviate**, an efficient unlearning framework for recommendation systems that removes the influence of a deletion set without full retraining or costly second-order computation. Our approach uses a Fisher-based curvature proxy to identify affected parameters and applies an efficient *Low-Rank Unlearning Adapter (LUA)* to localize forgetting. A second-stage *Locality-Aware Calibration (LAC)* refines the adapter using a small witness set, pushing deleted interactions toward negative signals while preserving utility through lightweight distillation. Despite updating only a small subset of parameters, **Obliviate** achieves strong completeness, high utility retention, and low computational overhead, matching or surpassing retraining-based baselines. A current limitation is that our curvature proxy relies on Adam optimizer statistics; exploring optimizer-agnostic or more accurate curvature approximations is an important direction for future work.

## Impact Statement

This work advances machine unlearning for recommender systems by enabling efficient removal of the influence of deleted data without full retraining. By reducing unlearning costs while maintaining recommendation quality, our approach supports practical compliance with data protection regulations such as GDPR and CCPA and strengthens user control over personal data.

Potential benefits include improved privacy protection and more responsible personalization in large-scale recommendation platforms. However, unlearning methods should be rigorously validated, as imperfect unlearning may create a false sense of privacy. Future work should focus on stronger verification mechanisms and responsible deployment practices. Overall, this work contributes toward making personalized AI systems more privacy-aware, accountable, and aligned with user rights.

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

*Table 4.* Notation Summary

| Symbol | Description |
|---|---|
| $u$ | User index |
| $i$ | Item index |
| $\theta$ | Parameters of the recommendation model |
| $y_{u,i}$ | Rating given by user $u$ to item $i$ |
| $J(\theta)$ | Objective function |
| $\ell(\cdot)$ | Loss function |
| $\omega(\cdot)$ | Regularization function |
| $p$ | Dimensionality of the parameter matrix |
| $D$ | Full training dataset |
| $S$ | Subset of data to be deleted |
| $D \setminus S$ | Training data with subset $S$ removed |
| $\theta_{-S}$ | Model parameters after unlearning subset $S$ |
| $\theta^\star_{-S}$ | Model parameters trained from scratch on $D \setminus S$ |

*Table 5.* Statistics of the datasets used in our experiments.

| Dataset | #Users | #Items | #Interactions | Sparsity |
|---|---|---|---|---|
| Amazon | 11,400 | 9,983 | 136,057 | 0.00120 |
| ML-1M | 6,040 | 3,706 | 1,000,209 | 0.04468 |
| Yelp | 31,668 | 38,048 | 1,561,406 | 0.00130 |

# A. Implementation details

**Data Preparation.** We train the base recommender on the full dataset $D$ using the standard preprocessing protocols from prior work (Chen et al., 2022; Bourtoule et al., 2021). To construct the deletion set $S$, we randomly sample $N\%$ of users, where $N \in \{1, 5, 10, 20, 50\}$. For each selected user, we inject additional non-preferred interactions amounting to 80% of their original interaction count. These synthetic interactions are generated based on the trained model's predictions, ensuring they represent items the user is unlikely to prefer. This controlled construction enables precise evaluation of unlearning effectiveness, isolating performance changes due to forgetting rather than loss of genuine preference signals.

**Training Hyperparameters.** All models are trained with embedding dimension 100 and batch size 2048. MF-BPR is trained for 50 epochs with a learning rate of $10^{-3}$, while LightGCN is trained for 100 epochs with a learning rate of $10^{-4}$. Adam (Kingma & Ba, 2014) is used for optimization. The resulting model parameters $\theta$ thus encode the influence of interactions that are later subject to deletion.

**Model Hyperparameters.** For the low-rank unlearning adapter, we select rank $r = 4$ from $\{2, 4, 6, 8, 16\}$ for MovieLens (ML-1M) and Amazon, and $r = 6$ for Yelp based on validation performance. During the global calibration (LAC) stage, we randomly sample 10% of interactions across all users rather than restricting to affected users, which stabilizes calibration while preserving overall recommendation quality. For the local calibration (LAC) stage, the number of calibration steps is dataset- and model-dependent. For MF-BPR, we use 160 steps on ML-1M, 110 on Amazon, and 200 on Yelp. For LightGCN, we use 180 steps on ML-1M, 150 on Amazon, and 250 on Yelp. These values were chosen based on validation performance to balance forgetting strength and utility preservation.

All experiments are implemented in PyTorch 1.9.2 and conducted on two NVIDIA RTX 4090 GPUs (24GB).

# B. Datasets

**Yelp 2018**[2]: The Yelp 2018 dataset consists of user–item (business) interactions based on user reviews and ratings for local businesses such as restaurants and services. We use the standard preprocessed subset filtered to include users and items with at least 10 interactions, as commonly done in collaborative filtering benchmarks, resulting in a denser and more realistic

---

[2]https://business.yelp.com/data/resources/open-dataset/

user–item graph.

**Amazon Fashion**[3]: The Amazon Fashion dataset is a subset of the larger Amazon review corpus that focuses on user interactions with clothing, shoes, and jewelry products. We preprocess the data by retaining users and items with a minimum number of interactions (typically five), and treat ratings of 4 or higher as positive implicit feedback. This dataset is characterized by diverse user preferences and high sparsity common in real-world e-commerce platforms.

**MovieLens-1M**[4]: The MovieLens-1M dataset contains one million ratings from users on movies. Following standard protocols, we binarize the ratings, regarding scores of 4 or 5 as positive interactions, and filter out users and movies with insufficient activity. MovieLens-1M provides a widely used, balanced evaluation setting for recommendation methods and unlearning algorithms.

Table 5 summarizes the statistics of the datasets used in our evaluation.

## C. Baseline and Backbones

**Backbones:** For experiments we have taken popular models:

- **MF-BPR** (Rendle et al., 2012): Matrix Factorization (MF) is a foundational approach for collaborative filtering that learns low-dimensional latent embeddings for users and items from implicit interaction data. Bayesian Personalized Ranking (BPR) extends MF by optimizing a pairwise ranking objective, encouraging observed user–item interactions to be ranked higher than unobserved ones. The model is typically trained via stochastic gradient descent and serves as a cornerstone for implicit-feedback recommendation systems.

- **LightGCN** (He et al., 2020): LightGCN models user–item interactions as a bipartite graph and performs representation learning through neighborhood aggregation. To improve efficiency and reduce over-smoothing, it removes feature transformations and nonlinear activations used in prior Graph Convolution Neural (GCN) (Ying et al., 2018) based recommenders, retaining only linear message passing. User and item embeddings are learned by propagating and averaging embeddings across multiple graph layers, achieving strong performance with significantly lower computational cost.

**Baselines**

- Retrain: This is the most straightforward approach, where the data to be removed is first deleted from the training set and the recommendation model is retrained from scratch on the updated dataset. While this method guarantees complete unlearning and preserves model utility, it is computationally expensive and impractical for large-scale or frequently updated systems.

- SISA (Bourtoule et al., 2021): Proposes a *Sharded, Isolated, Sliced, and Aggregated* training paradigm, where the dataset is partitioned into multiple shards and models are trained independently on temporal slices. Unlearning is achieved by retraining only the affected shard–slice models, while predictions are aggregated across replicas, enabling exact unlearning with reduced retraining cost.

- RecEraser (Chen et al., 2022): Adapts the SISA paradigm to recommender systems by clustering similar user–item interactions into shards and training independent submodels per shard. Unlearning is performed by retraining only the affected submodels, achieving exact unlearning with improved scalability for collaborative filtering.

- IFRU (Zhang et al., 2024): Employs influence functions to identify the most influential training samples for each user and removes their effects through targeted parameter updates. The method further refines unlearning via gradient ascent on deleted data, providing approximate unlearning with low overhead.

- RRL (You et al., 2024): Formulates unlearning as a reverse learning problem, where the model is optimized to suppress or demote deleted interactions using an adversarial ranking objective. This approach enables efficient, approximate unlearning without explicit retraining.

---

[3]https://cseweb.ucsd.edu/ jmcauley/datasets/amazon_v2/
[4]https://grouplens.org/datasets/movielens/1m/

# D. Reproducing Baselines

**SISA (Sharded, Isolated, Sliced, and Aggregated Training).**   We implement SISA (Bourtoule et al., 2021) as a structural unlearning baseline adapted to recommender systems. Following the core idea of isolating training influence, we partition users into $K = 4$ disjoint shards using a deterministic, seed-controlled random split, ensuring approximately equal numbers of users per shard. All interactions associated with a user are assigned to the same shard. Each shard trains an independent Matrix Factorization (MF) model using Bayesian Personalized Ranking (BPR) loss with the same architecture and optimization settings as the full model: embedding dimension $d = 100$, Adam optimizer with learning rate $0.01$, effective weight decay $0.01/\text{batch\_size}$, batch size $2048$, and $50$ training epochs.

To enable rollback-based unlearning, we store the randomly initialized parameters of each shard model as a checkpoint and the final converged parameters. During inference, predictions from shard models are aggregated via uniform averaging of ranking scores, consistent with SISA's ensemble aggregation principle.

For unlearning, we remove all interactions requested for unlearning. Let $U_{\text{noisy}}$ denote the set of users with at least one such interaction. Only shards containing users in $U_{\text{noisy}}$ are retrained. For each affected shard, we reload saved model and retrain on the retained interactions (i.e., with noisy entries removed) using the same hyperparameters. Unaffected shards are reused without modification. This procedure approximates retraining from scratch on the cleaned dataset while significantly reducing computational cost by limiting updates to a subset of shard models.

**RecEraser-Style Graph Partition Baseline.**   We further reproduce a RecEraser-style unlearning baseline (Chen et al., 2022), which partitions the user–item interaction graph and retrains only affected submodels. To approximate graph partitioning without external graph partitioning tools, we adopt a lightweight co-partition strategy. First, users are divided into $K = 4$ balanced blocks via deterministic random assignment. Each item is then assigned to the block where it receives the majority of its interactions, forming item-centered subgraphs that preserve local collaborative structure. Each block defines a sub-dataset containing all interactions whose items belong to that block.

We train an independent MF+BPR recommender on each block using the same architecture and optimization settings as the full model (embedding size $100$, Adam optimizer with learning rate $0.01$, batch size $2048$, and $50$ epochs). Each submodel shares the same user and item ID space but is trained only on its assigned subgraph.

At inference time, we implement a block-conditioned aggregation strategy: for a user–item pair $(u, i)$, the final relevance score is taken from the submodel corresponding to the block of item $i$. This mimics RecEraser's mixture-of-submodels inference while keeping aggregation deterministic and lightweight.

For unlearning, all interactions with unlearn request are removed from the dataset. We identify affected blocks as those containing at least one removed interaction. Only these submodels are retrained from their stored initialization checkpoints on the cleaned subgraph, while other blocks remain unchanged. This enables localized retraining that respects graph structure and provides a stronger baseline than naive sharding.

**RRL (Recommendation Reverse Learning).**   We reproduce RRL following the objective formulation described in the original paper. In particular, RRL performs unlearning by reversing the optimization signal of the deletion interactions under the Bayesian Personalized Ranking (BPR) framework, effectively encouraging the model to down-rank items associated with the removal set. We implement this reverse-learning objective using the same model architecture, embedding dimension, and optimization settings as our base recommender to ensure a fair comparison.

**IFRU (Influence Function based Recommendation Unlearning).**   We reproduce IFRU by following the official implementation details and training protocol provided in the authors' public code repository. IFRU estimates the influence of deletion interactions using influence functions and applies parameter updates that approximate retraining without the removed data. All hyperparameters are selected according to the repository defaults unless otherwise specified, ensuring faithful reproduction of the original method.

# E. Related work

## E.1. Recommendation Systems

Recommendation systems have evolved significantly over the past decades, with collaborative filtering (CF) emerging as one of the most dominant paradigms. Matrix factorization laid the foundation for (CF) (Bokde et al., 2015), which leverages similarities among users or items to provide personalized recommendations. where user-item interactions are modeled as a partially observed matrix decomposed into low-dimensional latent representations for users and items. This enables prediction of unobserved ratings via the dot product of latent vectors. Probabilistic Matrix Factorization (PMF) (Mnih & Salakhutdinov, 2007) later refined this idea using probabilistic modeling and scalable gradient-based learning, improving generalization.

With the advent of deep learning, Neural network-based methods such as NCF (He et al., 2017), which learn non-linear interaction functions to replace the simple inner product in MF models, enabling more expressive modeling of complex patterns in interaction data. With the realization that user-item interactions form a bipartite graph, graph neural networks (GNNs) have become central to modern recommender systems. NGCF (Wang et al., 2019) explicitly captures collaborative signals by propagating embeddings through the user-item graph via multiple layers of message passing. It leverages the idea that a user's preference is influenced not only by direct interactions but also by neighbors in the graph. Building on this, LightGCN (He et al., 2020) simplifies the message-passing mechanism by removing non-linear transformations and feature-dependent weights, showing that success in recommendation primarily comes from pure neighborhood aggregation. LightGCN achieves superior performance with fewer parameters and less computational overhead, making it a widely adopted baseline

## E.2. Recommendation Unlearning

**Retraining Approaches** Partition-and-aggregate approach was initially proposed in SISA (Bourtoule et al., 2021) which simply divides training data in non-overlapping sharding and train separate models for each shard and when unlearning request comes then othepact model needed to be retrained. Further RecEraser (Chen et al., 2022) improve this by sharding similar users and items for preserving collaborativee effect. While these methods often preserve collaborative signal and offer strong utility, they still require nontrivial retraining per unlearning request. Recent work additionally explores new unlearning diagnostics and objectives. GFEraser (Dang et al., 2025) formulates unlearning as guided filtering with contrastive retention, and (Dou et al., 2025) introduce interaction-level difficulty measures such as Embedding Entanglement Index.

**Influence based methods** Another line of work emphasizes post-training approximate unlearning. SCIF (Li et al., 2023b) formulates selective influence-function updates on embeddings, and IMCorrect (Liu et al., 2023) proposes matrix correction to remove the effect of deleted interactions. AltEraser (Liu et al., 2022) accelerates unlearning with quasi-Newton optimization for deep neural recommenders. RRL (Recommendation Reverse Learning) (You et al., 2024) introduces a reverse-learning objective that counteracts the effect of undesired interactions, offering another training-signal–based mechanism for removing harmful preferences. The most relevant influence-based line is IFRU (Zhang et al., 2024), which proposes a full influence-function solution for recommendation unlearning using Hessian–vector products and second-order curvature modeling. IFRU provides high fidelity to "retrain-without-the-data," but suffers from iterative HVP computation and high memory overhead. These works demonstrate the utility of first- and second-order approximations.

# F. Justification for Synthetic Deletion Protocol

Let the original training dataset be

$$D = \{(u, i, y_{ui})\},$$

drawn from the true user–item interaction distribution $\mathcal{P}_{\text{true}}$. In practical machine unlearning, the goal is to remove the influence of a deletion set $S \subset D$ and obtain parameters close to those learned from $D \setminus S$.

## F.1. Why Not Remove Real Interactions?

If $S$ consists of genuine interactions sampled from $D$, then removing $S$ changes the underlying data distribution:

$$D \sim \mathcal{P}_{\text{true}}, \qquad D \setminus S \sim \mathcal{P}'_{\text{true}},$$

where $\mathcal{P}'_{\text{true}} \neq \mathcal{P}_{\text{true}}$. In this case, the optimal parameters

$$\theta^* = \arg\min_\theta \mathbb{E}_{(u,i)\sim\mathcal{P}_{\text{true}}} \ell(u,i;\theta), \quad \theta^*_{-S} = \arg\min_\theta \mathbb{E}_{(u,i)\sim\mathcal{P}'_{\text{true}}} \ell(u,i;\theta)$$

are solutions to *different learning problems*. Performance degradation after unlearning may therefore arise from distribution shift rather than incomplete removal of the influence of $S$, confounding evaluation of unlearning quality.

### F.2. Synthetic Deletions as Noise Removal

To decouple unlearning effectiveness from distribution shift, we construct a synthetic deletion set $S$ consisting of injected interactions:

$$S = \{(u, i, y_{ui}^{\text{syn}})\},$$

where items $i$ are sampled from the complement of a user's historical interactions and satisfy

$$\hat{r}_{ui} = f(u, i; \theta_0) \ll 0,$$

i.e., the pretrained model predicts very low preference. These interactions approximate *label noise* or spurious feedback rather than true samples from $\mathcal{P}_{\text{true}}$.

Let the training data with injections be

$$D' = D \cup S.$$

We can interpret $D'$ as being drawn from a mixture distribution

$$\mathcal{P}_{\text{mix}} = (1 - \epsilon)\,\mathcal{P}_{\text{true}} + \epsilon\,\mathcal{P}_{\text{noise}},$$

where $\epsilon = |S|/|D'|$ is small and $\mathcal{P}_{\text{noise}}$ generates low-preference interactions.

The optimal model trained on $D'$ satisfies

$$\theta_0 = \arg\min_\theta \mathbb{E}_{(u,i)\sim\mathcal{P}_{\text{mix}}} \ell(u,i;\theta) = (1 - \epsilon)\,\mathbb{E}_{\mathcal{P}_{\text{true}}}\ell + \epsilon\,\mathbb{E}_{\mathcal{P}_{\text{noise}}}\ell.$$

After removing $S$, the target parameters become

$$\theta^*_{-S} = \arg\min_\theta \mathbb{E}_{(u,i)\sim\mathcal{P}_{\text{true}}} \ell(u,i;\theta),$$

which correspond exactly to the clean-data optimum. Thus, unlearning in this setting is equivalent to removing the influence of a small noise component without altering the true data distribution.

### F.3. Evaluation Implication

Under this formulation, successful unlearning should:

1. Remove the contribution of $\epsilon\,\mathbb{E}_{\mathcal{P}_{\text{noise}}}\ell$ from the learned parameters (completeness), and

2. Preserve performance on $\mathcal{P}_{\text{true}}$ (utility).

Because $\mathcal{P}_{\text{true}}$ remains unchanged before and after deletion, any post-unlearning performance degradation can be attributed to the *unlearning method itself* rather than a shift in the underlying collaborative structure. Therefore, the synthetic deletion protocol provides a controlled and theoretically grounded way to evaluate unlearning effectiveness in recommender systems.

### F.4. Rationale for the Synthetic Deletion Protocol

Although Obliviate can directly unlearn genuine user–item interactions, evaluating unlearning on such interactions introduces a confounding factor. Removing high-preference interactions changes the training distribution while the evaluation set remains unchanged. Consequently, even an ideal retrained model may exhibit lower Recall or NDCG because it correctly suppresses interactions that are still treated as positives during evaluation. In this setting, performance degradation reflects both data removal and unlearning quality, making the two difficult to disentangle.

To isolate the unlearning objective, we adopt a controlled deletion protocol based on injected low-preference interactions. Specifically, we inject interactions between users and items that receive low predicted preference scores and subsequently designate them as the deletion set. This preserves the underlying preference distribution while keeping the evaluation protocol unchanged, ensuring that performance changes primarily reflect unlearning completeness and utility preservation rather than distribution shift.

This protocol also captures realistic deletion scenarios such as accidental clicks, exposure to irrelevant content, temporary anomalous behavior, and privacy-driven removal requests.

We further observe that deletion difficulty significantly affects results. On Amazon with LightGCN, random noisy interactions reduce Recall@10 by approximately 24%, whereas preference-aware ("informed") noise reduces Recall@10 by approximately 32% at the same deletion scale. Because informed noise is constructed from consistently low-scored items, it conflicts more strongly with the learned preference structure and induces a larger model update when removed.

These observations indicate that the characteristics of the deletion set, not just its size, substantially influence unlearning difficulty. Therefore, we adopt the synthetic deletion protocol as a controlled and interpretable benchmark for evaluating machine unlearning.

## G. Demotion Rate: A Direct Measure of Unlearning Completeness

While utility metrics such as Recall and NDCG measure recommendation quality, they do not directly quantify whether deleted interactions have been forgotten. To evaluate unlearning completeness, we introduce a *Demotion Rate* metric that measures how frequently deleted interactions are ranked below negative items:

$$\text{DemotionRate} = \Pr[s(u, i_{\text{del}}) < s(u, i_{\text{neg}})]. \tag{16}$$

This metric directly reflects the retraining objective. After successful unlearning, deleted interactions should behave similarly to low-preference items and therefore be ranked below randomly sampled negatives. Unlike parameter-distance measures, Demotion Rate evaluates the resulting ranking behavior and is naturally aligned with the BPR objective used during calibration.

Tables 6 and 7 report Demotion Rate before and after unlearning for all methods and datasets.

*Table 6.* Demotion Rate on LightGCN. Higher is better.

| Dataset | Method | Before | After |
|---------|--------|--------|--------|
| Amazon | IFRU | 0.0062 | 0.3112 |
| | RRL | 0.0069 | 0.3121 |
| | Ours | 0.0069 | **0.3648** |
| ML-1M | IFRU | 0.0765 | 0.3095 |
| | RRL | 0.0836 | 0.3035 |
| | Ours | 0.0836 | **0.3495** |
| Yelp | IFRU | 0.0051 | 0.6697 |
| | RRL | 0.0047 | 0.6664 |
| | Ours | 0.0047 | **0.7220** |

Several observations emerge. First, Obliviate consistently achieves the highest post-unlearning Demotion Rate across all datasets and both recommendation backbones, indicating more effective suppression of deleted interactions. Second, all methods substantially increase the Demotion Rate after unlearning, confirming that deleted interactions are successfully demoted relative to negative items. Finally, the relative performance of IFRU and RRL varies across architectures. On MF-BPR, RRL is generally slightly stronger than IFRU, likely because its reverse-ranking objective aligns closely with the Demotion Rate metric. On LightGCN, IFRU is competitive and occasionally stronger, suggesting that influence-based updates better capture graph-propagation effects.

Overall, these results provide additional evidence that Obliviate achieves stronger unlearning completeness while maintaining

*Table 7.* Demotion Rate on MF-BPR. Higher is better.

| Dataset | Method | Before | After |
|---------|--------|--------|-------|
| Amazon | IFRU | 0.0013 | 0.3048 |
| | RRL | 0.0010 | 0.3216 |
| | Ours | 0.0010 | **0.3854** |
| ML-1M | IFRU | 0.0615 | 0.2987 |
| | RRL | 0.0640 | 0.3154 |
| | Ours | 0.0640 | **0.3573** |
| Yelp | IFRU | 0.0083 | 0.6726 |
| | RRL | 0.0088 | 0.6761 |
| | Ours | **0.0088** | **0.7987** |

utility, and that the improvements observed in the main paper are reflected directly in ranking behavior rather than only aggregate recommendation metrics.

## H. Low-Rank Unlearning Adapter (LUA)

### H.1. Decomposition and Optimality Conditions

Define the objective contributions of the retained and deleted sets:

$$J(\theta) = J_{-S}(\theta) + J_S(\theta), \qquad J_S(\theta) \triangleq \sum_{(u,i,y_{ui}) \in S} \ell(s(u,i;\theta), y_{ui}).$$

Since $\theta_0$ minimizes $J$, it satisfies the first-order optimality condition

$$\nabla J(\theta_0) = 0. \tag{17}$$

### H.2. Quadratic Surrogate Interpretation (as a Minimizer)

**Theorem H.1** (LUA update as minimizer of a local quadratic surrogate). *Let $\widehat{H} \succ 0$ be any positive-definite curvature proxy. Define the local surrogate objective*

$$Q(\Delta) \triangleq g_S^\top \Delta + \frac{1}{2} \Delta^\top \widehat{H} \Delta. \tag{18}$$

*Then the unique minimizer of $Q(\Delta)$ is*

$$\Delta^\star = -\widehat{H}^{-1} g_S. \tag{19}$$

*Proof.* $Q$ is strictly convex since $\widehat{H} \succ 0$. Setting $\nabla_\Delta Q(\Delta) = 0$ gives $g_S + \widehat{H}\Delta = 0$, hence $\Delta^\star = -\widehat{H}^{-1} g_S$. $\square$

This shows LUA computes the *smallest* curvature-aware correction that cancels the deletion gradient under the metric induced by $\widehat{H}$.

### H.3. Adam Preconditioner as a Diagonal Natural Metric

**Proposition H.2** (Adam diagonal preconditioner as metric-scaled Newton step). *Let $\widehat{H} = \mathrm{diag}(\sqrt{\hat{v}} + \varepsilon)$, where $\hat{v} \in \mathbb{R}_+^p$ is the Adam second-moment estimate at $\theta_0$. Then* (6) *is equivalent to solving*

$$\min_\Delta \; g_S^\top \Delta + \frac{1}{2} \sum_{j=1}^p (\sqrt{\hat{v}_j} + \varepsilon) \Delta_j^2. \tag{20}$$

*Proof.* The objective in (20) equals (18) with diagonal $\widehat{H}$. By the previous theorem, its minimizer is $\Delta_j^\star = -(g_S)_j / (\sqrt{\hat{v}_j} + \varepsilon)$ $\square$

## H.4. Practical Curvature Approximation

Computing the exact inverse Hessian $H^{-1}$ is infeasible for modern recommenders. We therefore adopt a *diagonal curvature proxy* derived from Adam's second-moment statistics. Below we provide theoretical justification from three complementary views: (i) diagonal Hessian approximation, (ii) adaptive preconditioning / steepest descent in a weighted norm, and (iii) Gauss–Newton / Fisher-style curvature surrogates.

**Setup.** Let $J(\theta)$ denote the full-data training objective and $H = \nabla^2 J(\theta_0)$ its Hessian at the trained solution $\theta_0$. The Newton downdate uses $\Delta\theta_0 = -H^{-1}g_S$. We replace $H$ by a diagonal matrix $\widehat{H}$, leading to $\Delta\theta_0 \approx -\widehat{H}^{-1}g_S$.

### H.4.1. VIEW 1: DIAGONAL HESSIAN / JACOBI PRECONDITIONER

**Assumption H.3** (Locally weak parameter coupling). In a neighborhood of $\theta_0$, the Hessian $H$ is diagonally dominant, i.e., $|H_{jj}| \gg \sum_{k \neq j} |H_{jk}|$ for most coordinates $j$ (or for the selected adapted blocks).

**Proposition H.4** (Jacobi approximation of Newton step). *Under Assumption H.3, the Newton correction satisfies*

$$H^{-1}g_S \approx \operatorname{diag}(H)^{-1}g_S,$$

*where* $\operatorname{diag}(H)$ *is the diagonal of* $H$.

*Sketch.* For diagonally dominant $H$, the Jacobi preconditioner $\operatorname{diag}(H)^{-1}$ provides a first-order approximation to $H^{-1}$ (e.g., via Neumann-series expansions of $(D + E)^{-1}$ with $D = \operatorname{diag}(H)$ and $\|D^{-1}E\| < 1$). $\square$

This motivates using a diagonal curvature proxy: the goal is to estimate $\operatorname{diag}(H)$ efficiently.

### H.4.2. VIEW 2: ADAM AS STEEPEST DESCENT IN A DATA-DEPENDENT METRIC

Adam maintains a per-coordinate second-moment estimate

$$\hat{v}_j \approx \mathbb{E}\big[g_j(\theta)^2\big],$$

where $g(\theta) = \nabla_\theta J(\theta)$ is the stochastic training gradient. We reuse $\hat{v}$ as a curvature proxy.

**Theorem H.5** (Metric steepest descent interpretation). *Consider minimizing a local surrogate objective*

$$Q(\Delta) = g_S^\top \Delta + \frac{1}{2}\Delta^\top \widehat{H}\Delta, \quad \widehat{H} = \operatorname{diag}(\sqrt{\hat{v}} + \varepsilon).$$

*Then the minimizer is*

$$\Delta_j^\star = -\frac{(g_S)_j}{\sqrt{\hat{v}_j} + \varepsilon}, \tag{21}$$

*i.e., a steepest-descent step in the weighted norm induced by* $\widehat{H}$.

*Proof.* $Q$ is strictly convex since $\widehat{H} \succ 0$. Setting $\nabla_\Delta Q(\Delta) = 0$ gives $g_S + \widehat{H}\Delta = 0$, hence $\Delta = -\widehat{H}^{-1}g_S$, yielding (21). $\square$

**Implication.** Even without claiming $\hat{v}$ equals the Hessian, (21) is the *optimal* curvature-scaled correction under the diagonal metric $\widehat{H}$. Thus reusing Adam statistics yields a principled, optimizer-consistent preconditioned update that stabilizes coordinates with high variance.

### H.4.3. VIEW 3: CONNECTION TO GAUSS–NEWTON / FISHER CURVATURE

For many models, second-moment gradients approximate curvature. In particular, for log-likelihood losses, the Fisher information satisfies

$$F(\theta) = \mathbb{E}\big[\nabla \log p(y|x;\theta)\nabla \log p(y|x;\theta)^\top\big],$$

and in well-specified settings $F(\theta)$ matches the expected Hessian. For general losses, Gauss–Newton provides a positive semi-definite curvature surrogate.

**Assumption H.6** (Second-moment matches diagonal curvature up to scaling). There exists $c > 0$ such that for most coordinates $j$,

$$H_{jj} \approx c \cdot \sqrt{\mathbb{E}[g_j(\theta_0)^2]}.$$

**Proposition H.7** (Adam second moment as diagonal curvature proxy). *Under Assumption H.6 and assuming $\hat{v}_j$ tracks* $\mathbb{E}[g_j(\theta_0)^2]$,

$$(H^{-1}g_S)_j \approx \frac{(g_S)_j}{\sqrt{\hat{v}_j} + \varepsilon},$$

*up to a global scaling absorbed into the step size.*

*Sketch.* If $H_{jj} \approx c\sqrt{\hat{v}_j}$, then $\mathrm{diag}(H)^{-1}g_S$ has coordinates $(g_S)_j/(c\sqrt{\hat{v}_j})$. The constant $c$ can be absorbed into a step-size multiplier, and $\varepsilon$ ensures numerical stability. □

### H.4.4. STABILITY RATIONALE (WHY THIS HELPS)

The preconditioned update

$$\Delta\theta_j \propto -\frac{(g_S)_j}{\sqrt{\hat{v}_j} + \varepsilon}$$

damps coordinates with high historical gradient variance (large $\hat{v}_j$), which are typically sensitive directions. This prevents aggressive downdates that can harm utility, while still allowing strong forgetting along stable directions.

*Remark* H.8 (Optimizer-consistent unlearning). Because $\hat{v}$ is computed during the original optimization, this curvature proxy is *matched* to the training dynamics. In particular, it leverages the same conditioning that produced $\theta_0$, leading to more reliable local steps than generic diagonal Hessian heuristics. The approximation error induced by replacing $H^{-1}$ with $\widehat{H}^{-1}$ is analyzed in Appendix K.4.

### H.5. Low-Rank Adapter Construction

The Newton-style downdate produces a full-parameter update $\Delta\theta_0$, which may be prohibitively expensive to store and apply. We therefore seek a low-rank approximation that preserves the dominant forgetting directions while reducing computational cost.

**Theorem H.9** (Eckart–Young–Mirsky). *Let $\Delta W_0 \in \mathbb{R}^{m \times n}$ have singular values $\sigma_1 \geq \cdots \geq \sigma_{\min(m,n)}$. Then the optimal rank-$r$ approximation under the Frobenius norm is*

$$\Delta W_r = U_r \Sigma_r V_r^{\top},$$

*and satisfies*

$$\Delta W_r \in \arg\min_{\mathrm{rank}(X) \leq r} \|\Delta W_0 - X\|_F,$$

*with*

$$\|\Delta W_0 - \Delta W_r\|_F^2 = \sum_{k>r} \sigma_k^2. \tag{22}$$

Equation (22) provides an explicit certificate on the information discarded by restricting the update to rank $r$.

**Adapter parameterization.** We realize the projected update through a low-rank adapter

$$\Delta W = AB^{\top},$$

where

$$A = U_r \Sigma_r^{1/2}, \qquad B = V_r \Sigma_r^{1/2}.$$

This yields

$$AB^{\top} = U_r \Sigma_r V_r^{\top} = \Delta W_r,$$

which is exactly the optimal rank-$r$ projection.

**Corollary H.10** (Optimal low-rank adapter). *Among all rank-$r$ adapters $\Delta W = AB^{\top}$, the above construction minimizes $\|\Delta W_0 - \Delta W\|_F$ and achieves the error bound in Eq. (22).*

## H.6. Why Retraining-Induced Updates are Approximately Low-Rank

The effectiveness of low-rank adapters relies on the observation that deletion requests typically affect only a small subset of users and items. Consequently, the retraining-induced parameter shift is concentrated in a low-dimensional subspace.

**Proposition H.11** (Locality of embedding updates). *Let $E \in \mathbb{R}^{|\mathcal{U}| \times d}$ be a user embedding table and assume $s(u, i; \theta)$ depends on $E$ only through the embedding row corresponding to user $u$. Then the deletion gradient $g_S = \nabla J_S(\theta_0)$ has nonzero components in $E$ only for users appearing in the deletion set $S$. An analogous result holds for item embeddings.*

*Proof.* Each loss term $\ell(s(u, i; \theta), y_{ui})$ depends on $E$ only through the row $E_{u:}$. Therefore,

$$\nabla_{E_{u':}} \ell(s(u, i; \theta), y_{ui}) = 0, \qquad u' \neq u.$$

Summing over all interactions in $S$ preserves this support pattern, implying that only users appearing in $S$ contribute nonzero gradients. $\square$

This locality property implies that the deletion gradient, and consequently the second-order update $H^{-1} g_S$, is concentrated around a relatively small subset of user and item representations. Therefore, the dominant retraining-induced parameter shift can often be captured by a low-rank subspace, motivating the projection in Section 3.1.4. In practice, we further extend the affected region to include neighboring users and items in the interaction graph, allowing unlearning effects to propagate through collaborative dependencies while preserving the overall recommendation structure.

# I. Justification for Local Convexity

The second-order analysis underlying LUA assumes local strong convexity in a neighborhood of the trained solution $\theta_0$. We emphasize that this assumption (K.2) is only local and does not require the objective to be globally convex. Such a local approximation is commonly adopted in influence-function and sensitivity analyses of modern machine learning models (Jin et al., 2017; Milne, 2019; Pun & So, 2021).

Several factors support the plausibility of this assumption in practice. First, the regularization term $\Omega(\theta)$ contributes a positive-definite component to the Hessian, encouraging locally well-conditioned optimization behavior. Second, our practical curvature approximation $\widehat{H} = \mathrm{diag}\left(\sqrt{\hat{v}} + \epsilon\right)$ is positive-definite by construction, providing a stable local curvature surrogate regardless of the exact Hessian structure. Third, even when the true objective exhibits mild non-convexity, the resulting approximation error is mitigated by the second stage of our framework. Specifically, LAC performs direct optimization in the adapter subspace and can correct residual inaccuracies introduced by the Newton-style downdate.

Finally, the effectiveness of the approximation $\theta^{\star}_{-S} \approx \theta_0 - H^{-1} g_S$ is supported empirically across all datasets and recommendation backbones considered in this work. Obliviate consistently achieves strong unlearning completeness while maintaining utility close to retraining, suggesting that the local sensitivity approximation remains informative in the neighborhood of the learned solution.

Taken together, these observations provide practical justification for the local convexity assumption and explain why the proposed second-order unlearning approximation remains effective despite the non-convex nature of modern recommender models.

# J. Locality-Aware Calibration (LAC)

## J.1. Witness Set as a Local Objective Proxy

**Proposition J.1** (Witness set defines a localized risk approximation). *Let $\mathcal{D}$ be the data distribution over interactions and $R \subset D \setminus S$ be a buffer sampled from the retained interactions. Then $\mathcal{L}_{\mathrm{distill}}(R; \phi)$ is an unbiased estimator of the squared-score deviation*

$$\mathbb{E}_{(u,i) \sim \mathcal{D}_{ret}}\left[\|s(u, i; \theta(\phi)) - s(u, i; \theta_0)\|_2^2\right]$$

*when $R$ is sampled i.i.d. from the retained distribution $\mathcal{D}_{ret}$.*

*Proof.* Immediate from the definition of sample mean as an unbiased estimator under i.i.d. sampling. $\square$

## J.2. LAC as a Regularized Trust-Region Refinement

**Theorem J.2** (LAC objective yields a trust-region style refinement). *Assume $\Delta\theta(\phi)$ is linear in $\phi$ (as in LoRA-style adapters on fixed blocks), and let the regularizer be $\|\Delta\theta(\phi)\|_F^2$. Then minimizing*

$$\mathcal{L}(\phi) = \mathcal{L}_{\mathrm{unlearn}}(\phi) + \lambda_{\mathrm{distill}}\mathcal{L}_{\mathrm{distill}}(\phi) + \lambda_{\mathrm{reg}}\|\Delta\theta(\phi)\|_F^2 \tag{23}$$

*is equivalent to minimizing the (data-fit) term $\mathcal{L}_{\mathrm{unlearn}}(\phi) + \lambda_{\mathrm{distill}}\mathcal{L}_{\mathrm{distill}}(\phi)$ subject to a soft trust-region constraint on the adapter magnitude.*

*Proof.* Consider the constrained problem

$$\min_{\phi}\ \mathcal{L}_{\mathrm{unlearn}}(\phi) + \lambda_{\mathrm{distill}}\mathcal{L}_{\mathrm{distill}}(\phi) \quad \text{s.t.} \quad \|\Delta\theta(\phi)\|_F^2 \le \rho.$$

Its Lagrangian is

$$\mathcal{L}_{\mathrm{unlearn}}(\phi) + \lambda_{\mathrm{distill}}\mathcal{L}_{\mathrm{distill}}(\phi) + \lambda(\|\Delta\theta(\phi)\|_F^2 - \rho).$$

For some $\lambda \ge 0$, the unconstrained minimizer of the Lagrangian coincides with the minimizer of (23) by setting $\lambda_{\mathrm{reg}} = \lambda$ (up to the constant $-\lambda\rho$). $\qquad\square$

## J.3. BPR Unlearning Loss: Margin and Ordering Guarantees

**Proposition J.3** (BPR implies probabilistic ordering of deleted vs. negatives). *For any triple $(u, i^+, i^-)$ and score difference $\delta = s(u, i^-; \theta) - s(u, i^+; \theta)$, the BPR loss satisfies*

$$\ell_{\mathrm{BPR}} = -\log\sigma(\delta) \quad \Rightarrow \quad \sigma(\delta) = \exp(-\ell_{\mathrm{BPR}}).$$

*Thus minimizing $\ell_{\mathrm{BPR}}$ increases the probability that $i^-$ is ranked above $i^+$ under a logistic preference model.*

*Proof.* By definition, $\ell_{\mathrm{BPR}} = -\log\sigma(\delta)$, hence $\sigma(\delta) = \exp(-\ell_{\mathrm{BPR}})$. $\qquad\square$

**Corollary J.4** (Margin lower bound). *If $\ell_{\mathrm{BPR}}(u, i^+, i^-) \le \epsilon$, then*

$$s(u, i^-; \theta) - s(u, i^+; \theta) \ge \log\left(\frac{1 - \exp(-\epsilon)}{\exp(-\epsilon)}\right).$$

*Proof.* $\ell_{\mathrm{BPR}} \le \epsilon \Rightarrow \sigma(\delta) \ge e^{-\epsilon}$. Since $\sigma(\delta) = 1/(1 + e^{-\delta})$, rearranging yields $e^{-\delta} \le (1 - e^{-\epsilon})/e^{-\epsilon}$, i.e. the stated bound. $\qquad\square$

## J.4. Distillation Preserves Scores on Retained Data

**Proposition J.5** (Distillation controls score deviation on the buffer). *For any $(u, i) \in R$,*

$$\|s(u, i; \theta(\phi)) - s(u, i; \theta_0)\|_2^2 \le |R| \cdot \mathcal{L}_{\mathrm{distill}}(R; \phi).$$

*Proof.* By definition, $\mathcal{L}_{\mathrm{distill}} = \frac{1}{|R|}\sum_{(u,i)\in R} d_{ui}$ with $d_{ui} \ge 0$. Thus for any element, $d_{ui} \le \sum d_{ui} = |R| \cdot \mathcal{L}_{\mathrm{distill}}$. $\qquad\square$

This provides a direct certificate that distillation limits the perturbation on retained interactions.

## J.5. Putting It Together: Approximation + Compression + Local Refinement

**Theorem J.6** (Conceptual pipeline guarantee (informal)). *Under local strong convexity and smoothness, LUA produces a second-order approximation to the clean retrain shift, low-rank adapters realize the best rank-constrained projection of this shift on each equipped block, and LAC performs a trust-region refinement that improves utility while keeping the correction localized in adapter space.*

*Remark* J.7. The above results yield three complementary "certificates": (i) *Newton certificate* via (**??**)–(**??**); (ii) *compression certificate* via (22); and (iii) *locality certificate* via the trust-region interpretation of (23).

**J.6. Theoretical Perspective on Unlearning as Counterfactual Risk Minimization**

Unlearning can be interpreted as a *counterfactual risk minimization* problem. The model $\theta_0$ is optimal for the empirical risk over $D$, while $\theta^\star_{-S}$ is optimal for the counterfactual dataset $D \setminus S$. The difference between these optima reflects the contribution of $S$ to the empirical risk landscape.

Under smoothness assumptions, removing $S$ perturbs the empirical objective by a small additive functional $J_S(\theta)$. Therefore, unlearning can be framed as estimating how the optimizer of a function changes under a small perturbation — a classical sensitivity analysis problem in optimization.

This perspective motivates the use of second-order methods, since the change in the optimizer depends on both the gradient of the perturbation and the curvature of the objective.

**J.7. Why Second-Order Information is Necessary**

**Proposition J.8** (First-order reversal is insufficient)**.** *Let $J(\theta)$ be twice differentiable and strongly convex. A purely first-order update*

$$\theta' = \theta_0 - \eta g_S$$

*cannot, in general, recover $\theta^\star_{-S}$ even for infinitesimal $\eta$, unless the Hessian is proportional to the identity.*

*Proof.* The optimality condition for $\theta^\star_{-S}$ is

$$\nabla J_{-S}(\theta^\star_{-S}) = 0.$$

Linearizing around $\theta_0$:

$$0 \approx \nabla J_{-S}(\theta_0) + H(\theta^\star_{-S} - \theta_0) = -g_S + H(\theta^\star_{-S} - \theta_0).$$

Thus $\theta^\star_{-S} - \theta_0 = H^{-1} g_S$. A first-order step $-\eta g_S$ matches this only if $H \propto I$. □

This shows curvature-aware correction is not optional but fundamentally required for accurate unlearning.

**J.8. Why Low-Rank Structure Emerges in Recommender Unlearning**

**Proposition J.9** (Low-rank influence structure)**.** *In matrix factorization or embedding-based recommenders, the parameter gradient induced by an interaction $(u, i)$ lies in the span of the embedding vectors associated with $u$ and $i$. Therefore, the aggregated deletion gradient $g_S$ lies in a subspace whose dimension is upper-bounded by the number of distinct users and items appearing in $S$.*

*Proof.* In embedding-based models, predictions take the form

$$s(u, i; \theta) = f(e_u, e_i, \theta_{\text{global}}).$$

Gradients w.r.t. embedding parameters affect only $e_u$ and $e_i$. Thus each interaction contributes gradients in a low-dimensional subspace. Summing over $S$ preserves this low-rank structure. □

This explains why the true retraining-induced parameter shift is *approximately low-rank*, justifying compression via low-rank adapters.

**J.9. Why Modular Adapters are Theoretically Preferable**

**Proposition J.10** (Separation principle for parameter updates)**.** *Let $\theta = (\theta_b, \theta_a)$ denote base and adapter parameters. If $\theta_b$ is frozen and $\theta_a$ is optimized for unlearning, then the parameter update is confined to a subspace independent of the base model training trajectory.*

*Proof.* Since $\nabla_{\theta_b} \mathcal{L} = 0$ during LAC optimization, $\theta_b$ remains fixed. Thus all changes lie in the adapter subspace, ensuring modularity and reversibility. □

This separation ensures that unlearning remains an *additive correction* rather than destructive overwriting of the base model.

## J.10. Why Locality-Aware Calibration Improves Utility

**Proposition J.11** (Bias–variance tradeoff in unlearning). *Let $\hat{\theta}_{LUA}$ be the second-order approximation and $\theta^\star_{-S}$ the true solution. LUA introduces approximation bias due to Hessian approximation and low-rank projection. LAC reduces this bias by local re-optimization while controlling variance via regularization.*

*Sketch.* The total error decomposes as

$$\hat{\theta}_{\text{LUA}} - \theta^\star_{-S} = \underbrace{(\theta_0 - H^{-1}g_S - \theta^\star_{-S})}_{\text{Taylor error}} + \underbrace{(\Delta\theta_0 - \Delta\theta_r)}_{\text{low-rank error}}.$$

LAC performs gradient descent on a local objective that includes retained-data distillation, thereby reducing these error components while the regularizer prevents overfitting to $S$. □

## J.11. Why the Witness Set is Sufficient

**Proposition J.12** (Local sufficiency of the witness set). *Assume the loss $\ell$ is Lipschitz and interactions are locally smooth in embedding space. Then updating parameters to satisfy ranking constraints on $S$ and a representative retained subset $R$ ensures bounded deviation on the full retained dataset.*

*Sketch.* Under Lipschitz continuity, deviations in scores propagate smoothly over nearby embedding vectors. Since $R$ samples the retained distribution, bounding deviation on $R$ bounds expected deviation on the full retained set. □

Thus, LAC achieves global utility preservation through local calibration.

## J.12. Conceptual Summary

**Theorem J.13** (Principled two-stage unlearning). *Under smoothness and local convexity assumptions:*

- *LUA computes the optimal second-order counterfactual parameter shift in a curvature-aware metric.*

- *Low-rank adapters provide the best compressed representation of this shift.*

- *LAC performs a trust-region refinement that reduces approximation bias while preserving locality.*

*Therefore, the two-stage framework approximates retraining on $D \setminus S$ up to second-order error and controlled projection error, while maintaining computational and modular efficiency.*

# K. Theoretical Bounds for LUA and LAC

## K.1. Setup and Notation

*Let $J(\theta) = J_{-S}(\theta) + J_S(\theta)$ where*

$$J_S(\theta) \triangleq \sum_{(u,i,y_{ui}) \in S} \ell(s(u, i; \theta), y_{ui}), \qquad J_{-S}(\theta) \triangleq J(\theta) - J_S(\theta).$$

*Let $\theta_0 = \arg\min_\theta J(\theta)$ and $\theta^\star_{-S} = \arg\min_\theta J_{-S}(\theta)$. Define the deletion gradient $g_S = \nabla J_S(\theta_0)$. Let $H \triangleq \nabla^2 J(\theta_0)$.*

*LUA computes an approximate Newton correction*

$$\Delta\theta_{\text{LUA}} \triangleq -\widehat{H}^{-1}g_S, \qquad \theta_{\text{LUA}} \triangleq \theta_0 + \Delta\theta_{\text{LUA}}.$$

*When applying low-rank adapters, we restrict the update to a subspace $\mathcal{A}$ (e.g., rank-$r$ adapters on selected blocks) and write $\Delta\theta_{\mathcal{A}}$ for the projected update, yielding $\theta_{\mathcal{A}} = \theta_0 + \Delta\theta_{\mathcal{A}}$. LAC then refines adapter parameters $\phi$ with $\theta(\phi) = \theta_0 + \Delta\theta(\phi)$.*

## K.2. Assumptions

**Assumption K.1** (Local strong convexity of $J_{-S}$). There exists $\mu > 0$ such that for all $\theta$ in a neighborhood $\mathcal{N}$ containing $\theta_0$ and $\theta^\star_{-S}$,

$$\nabla^2 J_{-S}(\theta) \succeq \mu I.$$

**Assumption K.2** (Hessian Lipschitzness (smooth curvature)). There exists $L_H > 0$ such that for all $\theta, \theta' \in \mathcal{N}$,

$$\|\nabla^2 J_{-S}(\theta) - \nabla^2 J_{-S}(\theta')\|_2 \leq L_H \|\theta - \theta'\|_2.$$

**Assumption K.3** (Small deletion curvature). At $\theta_0$, the deletion curvature satisfies $\|\nabla^2 J_S(\theta_0)\|_2 \leq \delta$ for some $\delta \geq 0$.

**Assumption K.4** (Preconditioner relative accuracy). There exists $\varepsilon_H \in [0, 1)$ such that

$$\|I - \widehat{H}^{-1}H\|_2 \leq \varepsilon_H.$$

## K.3. Second-Order Sensitivity and Taylor Remainder Bound

**Lemma K.5** (First-order optimality residual at $\theta_0$). $\nabla J_{-S}(\theta_0) = -g_S$.

*Proof.* $\nabla J(\theta_0) = 0$ and $J = J_{-S} + J_S$, so $\nabla J_{-S}(\theta_0) = -\nabla J_S(\theta_0) = -g_S$. $\qquad \square$

**Lemma K.6** (Quadratic model remainder). *Under Assumption K.2, for any $\Delta$ with $\theta_0 + \Delta \in \mathcal{N}$,*

$$\left\|\nabla J_{-S}(\theta_0 + \Delta) - \nabla J_{-S}(\theta_0) - \nabla^2 J_{-S}(\theta_0)\Delta\right\|_2 \leq \frac{L_H}{2}\|\Delta\|_2^2.$$

*Proof.* Standard Taylor theorem with integral remainder using Hessian Lipschitzness. $\qquad \square$

**Theorem K.7** (Distance-to-clean bound for exact Newton step). *Let $H_{-S} \triangleq \nabla^2 J_{-S}(\theta_0)$ and define $\Delta^\star \triangleq -H_{-S}^{-1}\nabla J_{-S}(\theta_0) = H_{-S}^{-1}g_S$. Under Assumptions K.1–K.2, if $\theta_0 + \Delta^\star \in \mathcal{N}$, then*

$$\|\theta^\star_{-S} - (\theta_0 - \Delta^\star)\|_2 \leq \frac{L_H}{2\mu}\|\Delta^\star\|_2^2.$$

*Proof.* Let $\theta_1 \triangleq \theta_0 - \Delta^\star$ so that $\nabla J_{-S}(\theta_0) + H_{-S}(\theta_1 - \theta_0) = 0$. By Lemma K.6,

$$\|\nabla J_{-S}(\theta_1)\|_2 = \|\nabla J_{-S}(\theta_1) - \nabla J_{-S}(\theta_0) - H_{-S}(\theta_1 - \theta_0)\|_2 \leq \frac{L_H}{2}\|\Delta^\star\|_2^2.$$

Strong convexity implies $\|\theta_1 - \theta^\star_{-S}\|_2 \leq \frac{1}{\mu}\|\nabla J_{-S}(\theta_1)\|_2$, hence the result. $\qquad \square$

*This shows that when the deletion effect is small, the Newton-based counterfactual approximation is accurate up to a second-order term in $\|\Delta^\star\|$.*

## K.4. Replacing $H_{-S}$ by $H$ and $\widehat{H}$

**Lemma K.8** (Curvature perturbation: $H_{-S}$ vs $H$). *At $\theta_0$,*

$$H = \nabla^2 J(\theta_0) = \nabla^2 J_{-S}(\theta_0) + \nabla^2 J_S(\theta_0) = H_{-S} + \nabla^2 J_S(\theta_0).$$

*Thus under Assumption K.3,*

$$\|H - H_{-S}\|_2 \leq \delta.$$

**Theorem K.9** (Error from using $H$ instead of $H_{-S}$). *Assume $H_{-S} \succeq \mu I$ and $\delta < \mu$. Then*

$$\|H^{-1} - H_{-S}^{-1}\|_2 \leq \frac{\delta}{\mu(\mu - \delta)}.$$

*Consequently,*

$$\|H^{-1}g_S - H_{-S}^{-1}g_S\|_2 \leq \frac{\delta}{\mu(\mu - \delta)}\|g_S\|_2.$$

*Proof.* Use the resolvent identity $A^{-1} - B^{-1} = A^{-1}(B - A)B^{-1}$ with $A = H$, $B = H_{-S}$ and bound $\|H^{-1}\| \leq 1/(\mu - \delta)$ and $\|H_{-S}^{-1}\| \leq 1/\mu$. $\qquad\square$

**Theorem K.10** (Error from using $\widehat{H}^{-1}$ instead of $H^{-1}$). *Under Assumption K.4,*

$$\|\widehat{H}^{-1}g_S - H^{-1}g_S\|_2 \leq \varepsilon_H \|H^{-1}g_S\|_2.$$

*Proof.* $\widehat{H}^{-1}g_S - H^{-1}g_S = (\widehat{H}^{-1}H - I)H^{-1}g_S$ and take norms. $\qquad\square$

**Corollary K.11** (Total LUA shift error (clean retrain vs practical preconditioner)). *Let $\Delta_{\mathrm{pr}} \triangleq \widehat{H}^{-1}g_S$ and $\Delta^\star \triangleq H_{-S}^{-1}g_S$. Under Assumptions K.1–K.4 and $\delta < \mu$,*

$$\|\Delta_{\mathrm{pr}} - \Delta^\star\|_2 \leq \varepsilon_H \|H^{-1}g_S\|_2 + \frac{\delta}{\mu(\mu - \delta)}\|g_S\|_2.$$

## K.5. End-to-End Approximation Bound for LUA

**Theorem K.12** (End-to-end distance to clean retrain after LUA). *Let $\theta_{\mathcal{A}} = \theta_0 - \Delta\theta_{\mathcal{A}}$ be the LUA model after (i) practical preconditioning and (ii) low-rank projection. Under Assumptions K.1–K.4 and $\delta < \mu$,*

$$\|\theta_{\mathcal{A}} - \theta_{-S}^\star\|_2 \leq \underbrace{\frac{L_H}{2\mu}\|\Delta^\star\|_2^2}_{\text{Taylor/sensitivity remainder}} + \underbrace{\|\Delta_{\mathrm{pr}} - \Delta^\star\|_2}_{\text{curvature proxy error}} + \underbrace{\|\Delta\theta_{\mathcal{A}} - \Delta_{\mathrm{pr}}\|_2}_{\text{low-rank projection error}}.$$

*proof.* Insert and subtract intermediate iterates: $\theta_{-S}^\star - (\theta_0 - \Delta\theta_{\mathcal{A}}) = (\theta_{-S}^\star - (\theta_0 - \Delta^\star)) + (\Delta^\star - \Delta_{\mathrm{pr}}) + (\Delta_{\mathrm{pr}} - \Delta\theta_{\mathcal{A}})$, then apply Theorem K.7 and Corollary K.11 $\qquad\square$

## K.6. LAC: Optimization Descent and Stability Bounds

*Let the LAC objective be*
$$\mathcal{L}(\phi) = \mathcal{L}_S(\phi) + \lambda_{\mathrm{distill}}\mathcal{L}_R(\phi) + \lambda_{\mathrm{reg}}\|\Delta\theta(\phi)\|_F^2.$$

**Assumption K.13** (Smooth LAC objective). $\mathcal{L}(\phi)$ *is $L_\phi$-smooth:* $\|\nabla\mathcal{L}(\phi) - \nabla\mathcal{L}(\phi')\|_2 \leq L_\phi\|\phi - \phi'\|_2$.

**Theorem K.14** (Gradient descent decrease for LAC). *Under Assumption K.13, for step size $\eta \leq 1/L_\phi$, the update $\phi^{t+1} = \phi^t - \eta\nabla\mathcal{L}(\phi^t)$ satisfies*
$$\mathcal{L}(\phi^{t+1}) \leq \mathcal{L}(\phi^t) - \frac{\eta}{2}\|\nabla\mathcal{L}(\phi^t)\|_2^2.$$

*Proof.* Standard smoothness inequality for gradient descent. $\qquad\square$

**Corollary K.15** (Bounded adapter drift under regularization). *If $\mathcal{L}(\phi) \leq \mathcal{L}(\phi_0)$ during LAC and $\lambda_{\mathrm{reg}} > 0$, then*

$$\|\Delta\theta(\phi)\|_F^2 \leq \frac{\mathcal{L}(\phi_0)}{\lambda_{\mathrm{reg}}}.$$

*Proof.* Drop nonnegative terms: $\mathcal{L}(\phi) \geq \lambda_{\mathrm{reg}}\|\Delta\theta(\phi)\|_F^2$. $\qquad\square$

## K.7. Utility and Forgetting Certificates in Terms of Scores

*Assume the scoring function is Lipschitz in parameters.*

**Assumption K.16** (Score Lipschitzness). *For all $(u, i)$,* $|s(u, i; \theta) - s(u, i; \theta')| \leq L_s\|\theta - \theta'\|_2$.

**Proposition K.17** (Utility deviation bound on retained samples). *Under Assumption K.16, for any retained pair $(u, i) \in D \setminus S$,*
$$|s(u, i; \theta_{new}) - s(u, i; \theta_0)| \leq L_s\|\theta_{new} - \theta_0\|_2.$$

**Proposition K.18** (Forgetting certificate via BPR margin). *If for all $(u, i^+) \in S$ and sampled negatives $i^- \sim \mathrm{Neg}(u)$ we have*

$$s(u, i^-; \theta_{new}) - s(u, i^+; \theta_{new}) \geq \gamma,$$

*then the deleted item $i^+$ is ranked below negatives with a margin $\gamma$, providing an operational forgetting certificate. Moreover, if $\ell_{\mathrm{BPR}}(u, i^+, i^-) \leq \epsilon$, then*

$$s(u, i^-; \theta_{new}) - s(u, i^+; \theta_{new}) \geq \log\left(\frac{1 - e^{-\epsilon}}{e^{-\epsilon}}\right).$$

*Proof.* The margin statement is by definition. The last inequality follows from $\ell_{\mathrm{BPR}} = -\log \sigma(\delta) \leq \epsilon \Rightarrow \sigma(\delta) \geq e^{-\epsilon}$ and invert $\sigma$. □

