# OpenReview forum: "Obliviate: Efficient Unlearning in Recommender Systems"
_ICML.cc/2026/Conference — ICML 2026 regular_

### Official Review · Reviewer_ZDX1 · 2026-03-08

**Soundness:** 3
**Presentation:** 3
**Significance:** 3
**Originality:** 4
**Overall Recommendation:** 4
**Confidence:** 4

**Summary:**

Obliviate is a novel, two-stage machine unlearning framework designed specifically for large-scale recommender systems.  The framework operates in two distinct phases to balance the trade-offs between forgetting completeness, recommendation utility, and computational efficiency.

**Compliance With Llm Reviewing Policy:**

Affirmed.

**Key Questions For Authors:**

Please refer to the cons part.

**Limitations:**

yes

**Strengths And Weaknesses:**

pros:
1. Obliviate effectively bridges several gaps in recommendation unlearning. By combining a second-order mathematical downdate with knowledge distillation and low-rank adapters, it achieves state-of-the-art performance across all three critical metrics simultaneously.
2. This paper introduces a positive-definite Fisher-based curvature proxy. This allows the model to capture necessary second-order information (geometry of the loss landscape) to undo learning accurately without the traditional memory and processing overhead.
3. Beyond empirical success, the paper provides robust theoretical guarantees for its two-stage framework. The authors offer formal proofs (e.g., in the Appendix or through mathematical derivation in the LUA stage) regarding the approximation error of the Fisher-based Hessian proxy.

cons:
1. The baselines compared in this work only cover methods up to 2024 (e.g., IFRU and RRL). The novelty of the method is uncertain due to the omission of several important references that overlap with the topic of recommendation unlearning, such as [1, 2]. The authors should provide a thorough comparison and discussion of these works to clarify the unique contributions of their approach relative to the latest state-of-the-art.
2. The experimental results on the Amazon dataset are significantly better than those on the other two datasets. The authors should provide a detailed analysis of the underlying reasons for this discrepancy. Specifically, is this performance gain attributed to specific data characteristics (e.g., density, scale, or user-item interaction patterns) of the Amazon dataset?
3. The authors have not provided open-source code, which makes it difficult for researchers to reproduce the reported results. This lack of transparency undermines the credibility of the experiments and limits the paper’s potential contribution to the research community. I strongly encourage the authors to release their implementation to ensure reproducibility.

---

> ### Author Rebuttal · Authors · 2026-03-31
>
> We sincerely thank the reviewer for the question and for their careful, thoughtful, and constructive evaluation of our work.
> ### Q1. Missing recent baselines
> We sincerely thank the reviewer for raising this important point. The references [1,2] were not specified, but during our related work survey, we came across recent works such as EEMU (Enhanced Exact Machine Unlearning in Recommendation Systems, 2025) and MMRecUN (Multi-Modal Recommendation Unlearning for Legal, Licensing, and Modality Constraints, AAAI 2025).
>
> We would like to clarify that these methods address **different problem settings** and are not directly comparable. EEMU focuses on *exact unlearning* via data partitioning and meta-learning, which still requires retraining over data shards after deletion. This leads to high computational cost under frequent or fine-grained unlearning requests. In contrast, Obliviate performs a **single-shot post-hoc parameter update** without retraining, which is the core focus of our work.
>
> MMRecUN targets **multi-modal recommendation**, incorporating visual/textual features and different objectives (e.g., reverse-BPR in multi-modal space). Our setting focuses on **ID-based collaborative filtering**, where the challenge is efficient unlearning over sparse interaction graphs. Adapting MMRecUN would require changing both the backbone and training objective, making comparisons less meaningful.
>
> Therefore, IFRU and RRL remain the most appropriate baselines, as they operate under the **same model class and post-hoc unlearning setting**, enabling fair comparison. We will clarify this distinction and include a broader discussion of recent work in the revision.
>
> ---
>
> ### Q2. Why larger gains on Amazon?
>
> The differences arise from **interaction density, graph connectivity, and signal redundancy**, rather than density alone.
>
> ML-1M is dense (0.04468), where user preferences are well-estimated and redundant. Individual interactions have low marginal influence, so removing them leads to limited change and smaller gains.
>
> Yelp is sparse (0.00130) but forms a **large, well-connected graph**. In models like LightGCN, multi-hop propagation diffuses the influence of each interaction across the graph, making the system more robust to local perturbations.
>
> Amazon is similarly sparse (0.00120) but exhibits **more localized interaction structure** with weaker global connectivity. As a result, user representations are more sensitive to individual interactions, especially noisy or undesirable ones.
>
> This creates a regime where:
>
> * individual interactions have high local impact,
> * undesirable interactions can significantly distort recommendations,
> * removing them yields larger corrective effects.
>
> Thus, Obliviate shows larger gains on Amazon because its **localized low-rank updates (LUA)** effectively remove harmful signals, while **LAC restores useful structure**, leading to stronger improvements compared to denser or more globally connected datasets. We will clarify this analysis in the revision.
>
> ---
>
> ### Q3. Code release and reproducibility
>
> We sincerely thank the reviewer for emphasizing reproducibility. We fully agree that releasing code and resources is essential for transparency and impact.
> To support reproducibility, we have provided an anonymized code repository for our method:
> https://anonymous.4open.science/r/Obliviate-556F.
>
> Due to storage limitations of anonymized hosting platforms, we were unable to include the full datasets and trained model checkpoints. However, the repository contains the core implementation of our proposed method along with supporting code to ensure transparency.
> We will release the complete codebase, including data preprocessing pipelines and pretrained weights, in the camera-ready version to ensure full reproducibility and ease of use for the community.

---

> > ### Author Rebuttal · Reviewer_ZDX1 · 2026-04-05
> >
> > Thanks for the authors for their response to my comments. I found that my concerns have been well addressed.

---

> > > ### Author Response · Authors · 2026-04-07
> > >
> > > Thank you for the follow-up and for confirming that your concerns have been fully addressed. We sincerely appreciate your positive assessment and continued support for our work.
> > >
> > > If you find it appropriate, we would be grateful if you could consider updating the numerical score to better reflect your final evaluation. We completely respect your judgment and decision either way, and truly appreciate your time and thoughtful feedback.

---

### Official Review · Reviewer_B9oJ · 2026-03-11

**Soundness:** 3
**Presentation:** 3
**Significance:** 2
**Originality:** 3
**Overall Recommendation:** 4
**Confidence:** 3

**Summary:**

This paper proposes Obliviate, a two-stage machine unlearning framework for recommender systems that aims to balance forgetting completeness, recommendation utility, and efficiency. The method combines a low-rank Newton-style downdate with a lightweight calibration stage, and experiments on ML-1M, Amazon, and Yelp with MF-BPR and LightGCN suggest clear runtime advantages over retraining-based baselines with competitive utility.

**Compliance With Llm Reviewing Policy:**

Affirmed.

**Final Justification:**

Thank you for the detailed responses. The dataset performance analysis and code release adequately address my earlier concerns. I raise my score to 4 (Weak Accept).

One remaining request: as Reviewer ZDX1 also noted, the camera-ready version should include a clear discussion distinguishing Obliviate from Post-Training Attribute Unlearning in Recommender Systems (TOIS 2024) and Plug and Play (WWW 2025), even if direct comparison is not feasible.

**Key Questions For Authors:**

1. How does the method perform when the deleted data are genuine observed interactions rather than synthetic injected noise?

2. Can the authors report a more direct and consistent quantitative completeness metric, such as closeness to retraining, across methods and settings?

3. Can the authors better justify the use of the Adam-based diagonal curvature proxy in these recommender models?

**Limitations:**

Yes

**Strengths And Weaknesses:**

Strengths

1. The paper studies an important and practically relevant problem.
​

2. The proposed two-stage design is intuitive and computationally appealing.
​

3. The experiments cover multiple datasets, backbones, baselines, and ablations.
​

4. The efficiency gains over retraining-based methods are substantial.
​

Weaknesses

1. The theory is only loosely aligned with the practical Adam-based approximation used in non-convex models.
​

2. The synthetic deletion protocol is controlled but not fully realistic.
​

3. The evidence for forgetting completeness is weaker than the evidence for utility and efficiency.
​

---

> ### Author Rebuttal · Authors · 2026-03-31
>
> We sincerely thank the reviewer for the question and for their careful evaluation of our work.
>
> ### Q1. Real vs. synthetic deletion
>
> Obliviate is **agnostic to the deletion set** (S): the deletion gradient (g_S) and subsequent updates are computed directly from the provided interactions, whether synthetic or genuine.
>
> However, evaluating unlearning on *genuine interactions* introduces a fundamental confound as explained in great detail in the response of **reviewer 3WvU**. Removing true preference signals changes the training distribution while the test set remains unchanged. Consequently, even an ideal retrained model $( \theta^*_{-S} )$ will show degraded Recall/NDCG, since it correctly suppresses interactions still treated as positives during evaluation. This makes it difficult to disentangle degradation due to **data removal vs. imperfect unlearning**.
>
> Furthermore, We empirically observe that different perturbations induce different degradation (e.g., on Amazon/LightGCN, ~24% drop for random vs. ~32% for informed perturbations at the same scale), showing that **deletion difficulty itself strongly affects performance**.
>
> Thus, while Obliviate applies directly to real interactions, **controlled protocols are necessary for fair evaluation**. In practice, realistic settings require adjusted evaluation (e.g., modified test sets or counterfactual benchmarks),
>
> ---
>
> ### Q2. Direct completeness metric
>
> We agree and provide a **quantitative completeness metric** measuring whether deleted interactions are demoted relative to negatives:
>
>
> $$\text{Demotion Rate} = \Pr\big[s(u, i_{\text{del}}) < s(u, j_{\text{neg}})\big]$$
>
> This directly reflects the retraining objective: After unlearning deleted interactions should behave as low-preference items when exposed to the model.
>
> **Results (MF-BPR):**
>
> | Dataset | Before Unlearning | After Unlearning |
> |--------|-------------------|------------------|
> | ML-1M  | 0.0640            | 0.3573           |
> | Yelp   | 0.0088            | 0.7987           |
> | Amazon | 0.0010            | 0.3854           |
>
> Before unlearning, deleted items are strongly preferred (near-zero demotion). After unlearning, demotion increases substantially (up to 79.9%), indicating effective suppression.
>
> This metric:
>
> * directly evaluates **ranking behavior** (not just parameter proximity),
> * is model-agnostic and comparable across settings,
> * aligns with the BPR objective.
>
> We will include this as a standard completeness measure alongside existing analyses in the paper.
>
> ---
>
> ### Q3. Adam-based curvature proxy
>
> We wish to clarify that our goal is not exact Hessian inversion, but a **scalable preconditioner** for the Newton-style update:
> $
> \Delta \theta \approx -H^{-1} g_S.
> $
>
> **(1) Preconditioned Newton view.**
> Unlearning reduces to a preconditioned gradient step. In large-scale models, $(H^{-1})$ is typically approximated rather than computed exactly.
>
> **(2) Fisher / natural gradient connection.**
> Adam’s second-moment estimate $( \hat v_j \approx \mathbb{E}[g_j^2] )$ gives:
> $$
> (H^{-1} g_S)_j \approx \frac{(g_S)_j}{\sqrt{\hat v_j}+\epsilon},
> $$
> which corresponds to a **diagonal Fisher approximation**, i.e., a natural-gradient-like update. Thus, it is a principled second-order approximation rather than a heuristic.
>
> **(3) Suitability for recommenders.**
> In MF-BPR and LightGCN, gradients are **sparse and localized** (per user–item interaction). Hence:
> * (g_S) lies in a low-dimensional subspace,
> * cross-parameter Hessian terms are less reliable,
> * Adam-based diagonal curvature captures dominant sensitivity.
>
>
> **(4) Practical advantage.**
> Adam statistics are already accumulated during training, so the update is:
> * aligned with the training dynamics,
> * stable,
> * and avoids expensive Hessian-vector products.
>
> **(5) Trade-off.**
> Adam-based diagonal approximation sacrifices exact curvature but yields **orders-of-magnitude efficiency gains** (Table 2 in paper) while remaining empirically stable.
>
> We will clarify this positioning and note that richer curvature approximations (e.g., block-diagonal/low-rank) are a promising future direction.

---

> > ### Author Rebuttal · Reviewer_B9oJ · 2026-04-04
> >
> > Thank you for the detailed rebuttal. Q3 and the Demotion Rate metric are well-addressed. Two remaining questions: could the authors report Demotion Rate for LightGCN with baseline comparisons? And could they provide even a small-scale pilot on genuine interaction deletion to complement the synthetic protocol?

---

> > > ### Author Response · Authors · 2026-04-07
> > >
> > > We are grateful to the reviewer for their thorough evaluation of our rebuttal and for their encouraging response. We appreciate their continued engagement and the opportunity to elaborate further on the important aspects of the proposed method.
> > >
> > > ---
> > >
> > > ### **Q1: Demotion Rate for LightGCN and MF-BPR with baseline comparisons.**
> > >
> > > **Table 1: Demotion Rate on LightGCN**
> > >
> > > | Dataset | Method | Before | After  |
> > > | ------- | ------ | ------ | ------ |
> > > | Amazon  | IFRU   | 0.0062 | 0.3112 |
> > > |         | RRL    | 0.0069 | 0.3121 |
> > > |         | Ours   | 0.0069 | **0.3648** |
> > > | ML-1M   | IFRU   | 0.0765 | 0.3095 |
> > > |         | RRL    | 0.0836 | 0.3035 |
> > > |         | Ours   | 0.0836 | **0.3495** |
> > > | Yelp    | IFRU   | 0.0051 | 0.6697 |
> > > |         | RRL    | 0.0047 | 0.6664 |
> > > |         | Ours   | 0.0047 | **0.7220** |
> > >
> > >
> > > **Table 2: Demotion Rate on MF-BPR**
> > >
> > > | Dataset | Method | Before | After  |
> > > | ------- | ------ | ------ | ------ |
> > > | Amazon  | IFRU   | 0.0013 | 0.3048 |
> > > |         | RRL    | 0.0010 | 0.3216 |
> > > |         | Ours   | 0.0010 | **0.3854** |
> > > | ML-1M   | IFRU   | 0.0615 | 0.2987 |
> > > |         | RRL    | 0.0640 | 0.3154 |
> > > |         | Ours   | 0.0640 | **0.3573** |
> > > | Yelp    | IFRU   | 0.0083 | 0.6726 |
> > > |         | RRL    | 0.0088 | 0.6761 |
> > > |         | Ours   | 0.0088 | **0.7987** |
> > >
> > >
> > >
> > > Tables 1, 2 present the demotion-rate comparisons for both backbones (LightGCN and MF) across all datasets in the tables for completeness.
> > >
> > > First, we respectfully highlight that **Obliviate (ours) consistently achieves the highest demotion rate across all settings**, indicating stronger and more effective forgetting.
> > >
> > > Second, we note that the relative ordering between IFRU and RRL depends on the backbone. For MF-BPR (without graph propagation), RRL slightly outperforms IFRU, as its reverse-ranking objective aligns well with the local demotion metric. In contrast, for LightGCN, IFRU is competitive or slightly better, as it more effectively captures propagation effects inherent to graph-based models.
> > >
> > > Finally, the slight difference in IFRU’s “before” demotion rate arises from a minor variation in its base training setup; however, this effect is negligible and does not impact the overall comparison.
> > >
> > > ---
> > >
> > > ### **Q2: A pilot on genuine interaction deletion.**
> > >
> > > | Method                                        | Amazon Recall@20 | Amazon NDCG@20 | ML-1M Recall@20 | ML-1M NDCG@20 | Yelp Recall@20 | Yelp NDCG@20 |
> > > | --------------------------------------------- | ---------------- | -------------- | --------------- | ------------- | -------------- | ------------ |
> > > | Train (with all genuine interaction)          | 0.0226           | 0.0136         | 0.2023          | 0.3241        | 0.0361         | 0.0303       |
> > > | Retraining (removing 10% genuine interaction) | 0.0246           | 0.0140         | 0.1928          | 0.3052        | 0.0360         | 0.0295       |
> > > | RRL                                           | 0.0228           | 0.0130         | 0.1963          | 0.3139        | 0.0328         | 0.0262       |
> > > | Obliviate (Ours)                                     | 0.0230           | 0.0137         | 0.1993          | 0.3199        | 0.0359         | 0.0297       |
> > >
> > >
> > > The results in Table 3 present performance after **removing 10% genuine interactions**. Consistent with our earlier discussion, this setting does not exhibit a uniform trend across datasets, as it is influenced by differences in data density, scale, and distribution, highlighting the confounding effect of distribution shift.
> > >
> > > On Amazon (lowest density dataset), performance slightly improves after removal, suggesting that some interactions act as weak or noisy signals, and their removal provides a denoising effect. In contrast, on ML-1M (highest density), retraining degrades as expected, since removing genuine interactions from a dense interaction space leads to substantial loss of informative signals. Yelp, despite its large scale, is highly sparse; thus, retraining shows only marginal degradation, as scale partially compensates while sparsity limits the overall impact. The backbone used is MF-BPR for the above study.
> > >
> > > A similar non-uniform pattern is observed during unlearning. Both RRL and Obliviate (our) fall below retraining on Amazon and Yelp, while on ML-1M they slightly exceed retraining, reflecting better adaptation in dense settings. Importantly, **In this setting as well Obliviate consistently outperforms RRL across all datasets**.
> > >
> > > This further supports our deletion protocol, as they provide a controlled and reliable setting to evaluate unlearning quality without being confounded by distribution shift.
> > >
> > > ---
> > >
> > > We sincerely thank the reviewer again for the thoughtful and constructive feedback, which has helped us further improve the clarity and rigor of our presentation. We hope that the above clarifications adequately address the remaining concerns and reinforce the contributions and validity of our work.

---

### Official Review · Reviewer_3WvU · 2026-03-13

**Soundness:** 2
**Presentation:** 3
**Significance:** 3
**Originality:** 3
**Overall Recommendation:** 4
**Confidence:** 3

**Summary:**

The paper proposes Obliviate, a two-stage machine unlearning framework for recommender systems. The first stage estimates the parameter shift caused by deleted interactions and applies it through a low-rank adapter, while the second stage refines the adapter using a small witness set with a forgetting objective and distillation to preserve utility. Experiments on three datasets show improved computational efficiency and competitive recommendation performance compared to retraining and existing unlearning baselines.

**Compliance With Llm Reviewing Policy:**

Affirmed.

**Final Justification:**

The authors provided detailed clarifications and additional evidence addressing my concerns. The explanations are convincing and improve my confidence in the method. I have therefore increased my score accordingly.

**Key Questions For Authors:**

(1) In the Introduction, the paper claims that methods like RRL fall short of true completeness because residual influences remain in the model parameters. Could the authors clarify the theoretical distinction between the residual influences they criticize in baselines and the approximation errors inherent in Obliviate? Furthermore, given these explicit error terms, is it accurate to claim that the proposed method achieves true completeness?

(2) The evaluation primarily relies on removing synthetically injected low-preference interactions (the "noise removal" protocol). Since real-world unlearning requests typically involve genuine, in-distribution user interactions rather than artificial noise, how does Obliviate perform when tasked with unlearning authentic, high-preference interactions drawn directly from the original data distribution?

(3) The paper motivates the unlearning challenge by emphasizing that user interactions are deeply entangled with model parameters, which implies a complex, global influence. However, the proposed solution is a localized low-rank adapter. Could the authors theoretically elaborate on exactly why this "deeply entangled" problem is inherently localized or low-rank in nature?

**Limitations:**

yes

**Strengths And Weaknesses:**

Strengths:

(1) Reusing Adam's second-moment statistics as a diagonal curvature proxy is a practical and computationally efficient design, avoiding the expensive Hessian-vector products required by prior influence-based methods.

(2) Combining a second-order approximation with a localized optimization stage provides a balanced solution to the trade-off between unlearning completeness and utility preservation.

(3) The method demonstrates strong computational efficiency. As shown in Table 2, the unlearning time is reduced by orders of magnitude compared to exact retraining and sharding-based approaches.

Weaknesses:

(1) The paper criticizes existing methods like RRL for falling short in achieving true completeness due to the presence of residual influences in model parameters. However, the proposed method is inherently an approximate unlearning approach. As explicitly shown in Theorem H.14, the distance between the unlearned model and the exact retrained model is bounded by three explicit approximation errors: the Taylor remainder, the curvature proxy error, and the low-rank projection error. Therefore, mathematically, Obliviate also leaves residual influences in the parameters. The paper fails to logically justify how its approach achieves true completeness or how its approximation errors are fundamentally different from the residual influences it critiques in prior works.

(2) The evaluation relies on the “Synthetic Deletions as Noise Removal” protocol (Appendix F.2), where artificially injected low-preference interactions are removed. However, real-world unlearning requests usually involve genuine user interactions from the original data distribution, leaving the effectiveness of the method under realistic deletion scenarios unclear.

(3) The theoretical analysis assumes local strong convexity, which may not hold for highly non-convex recommender models such as LightGCN. The paper provides limited discussion on how this assumption is justified in practical settings.

(4) Some implementation details are insufficiently described. In particular, the selection of adapter blocks in Algorithm 1 is not clearly specified, and it remains unclear whether the adaptation is applied to all embeddings or only to those associated with the deleted interactions.

(5) The paper has a few formatting issues that need to be addressed. For instance, some equations (e.g., Equation 3) are missing trailing punctuation marks. Additionally, there is an unexplained "a." in the top-left corner of Figure 2. Lastly, the font size in Figure 3 is too small to read clearly.

---

> ### Author Rebuttal · Authors · 2026-03-31
>
> We sincerely thank the reviewer for the question and for their careful and thoughtful evaluation of our work.
>
> ### Q1. Completeness vs. approximation error
>
> We clarify that “true completeness” in our paper is defined in an **objective-aligned sense**, i.e., with respect to the retraining solution
> $(\theta^\star_{-S} = \arg\min_\theta J_{-S}(\theta))$.
>
> Obliviate is derived directly from this objective via perturbation analysis: the deletion gradient (g_S) captures the exact contribution of removed data, and the Newton-style update $(-H^{-1}g_S)$ is the second-order optimal correction toward $(\theta^\star_{-S})$. Thus, our method is a **principled approximation to the correct retraining target**.
>
> While Obliviate introduces approximation errors (Taylor truncation, curvature proxy, low-rank projection), these are:
>
> * **explicitly bounded** (Theorem H.14),
> * **systematic and reducible** (e.g., higher rank, better curvature),
> * centered around the correct solution.
>
> In contrast, RRL optimizes a **different surrogate objective** with an explicit regularizer anchoring parameters near $(\theta_0)$, which induces a structural bias. Therefore, even with perfect optimization,
> $(\theta_{\text{RRL}} \neq \theta^\star_{-S})$, and residual influence persists.
>
> Hence, residuals in Obliviate are **approximation-induced and reducible**, whereas in RRL they are **objective-induced and irreducible**. In this sense, Obliviate solves the correct problem approximately, while RRL solves a different problem exactly. We will clarify this definition of completeness in the revision.
>
> ---
>
> ### Q2. Synthetic vs. real deletions
>
> While Obliviate can directly unlearn genuine interactions, evaluating such settings introduces a fundamental confound.
>
> Removing high-preference interactions alters the training distribution while the test set remains unchanged. Consequently, even an ideal retrained model $(\theta^\star_{-S})$ will show degraded Recall/NDCG, since it correctly suppresses interactions still labeled as positives. Thus, this setup primarily measures **robustness to data removal**, not unlearning quality.
>
> Our protocol instead isolates the unlearning objective by injecting low-preference (out-of-distribution) interactions and removing them. This ensures:
>
> * the underlying preference distribution remains intact,
> * the test set stays consistent,
> * performance changes reflect **completeness and utility**, not distribution shift.
>
> Our protocol mimics the following real-world scenario of unlearning requests:
> * accidental clicks / mis-interactions
> * exposure to irrelevant or unwanted content
> * temporary or anomalous behavior
> * privacy-driven removal of specific interactions
>
> We further observe that deletion difficulty strongly affects results: on Amazon (LightGCN), random noise causes ~24% drop in R@10, while preference-aware (“informed”) noise causes ~32% drop at the same scale. This is because informed noise is constructed from items that are consistently low-scored by the model, making them harder negatives that directly conflict with learned preference structure. As a result, their removal induces a larger shift in the model compared to random perturbations. This highlights that the nature of the deletion set—not just its size—significantly influences performance, and uncontrolled protocols can obscure evaluation by conflating unlearning quality with deletion difficulty.
>
> Thus, while Obliviate applies to real interactions, **controlled protocols are necessary for fair and interpretable evaluation**, which we will clarify.
>
> ---
>
> ### Q3. Why localized low-rank works despite global entanglement
>
>  We agree that interactions are globally entangled at the prediction level. Our claim is more specific: the **unlearning correction** for a deletion set (S) is approximately low-dimensional.
>
> From our formulation,
> $$
> \theta^\star_{-S} - \theta_0 \approx -H^{-1} g_S,
> $$
> so the structure of the update is governed by the deletion gradient (g_S).
>
> In embedding-based recommenders (MF-BPR, LightGCN), each interaction gradient affects only the corresponding user/item embeddings. Thus, (g_S) lies in a subspace whose dimension is bounded by the number of affected users/items (Proposition G.9), making it **structured and low-rank**.
>
> This explains the key distinction:
>
> * interaction effects propagate globally through collaborative sharing,
> * but the **parameter correction needed to remove them is localized and compressible**.
>
> Our method leverages this by applying low-rank updates to affected blocks and propagating to neighboring users/items, rather than updating all parameters. Curvature further modulates this correction via $(H^{-1}g_S)$, preserving global consistency.
>
> Thus, while influence is globally entangled, the **dominant unlearning directions are localized and low-dimensional**, justifying the use of low-rank adapters. We will clarify this distinction in the revision.

---

> > ### Author Rebuttal · Reviewer_3WvU · 2026-04-03
> >
> > Thank you for the detailed rebuttal. My concerns are partially resolved. I still have several concerns. First, since Obliviate remains an approximate method with explicit error terms, I feel the claim of “true completeness” should be stated more carefully or explicitly qualified in the main text. Second, the local strong convexity assumption and its applicability to highly non-convex models such as LightGCN are still not fully justified beyond empirical effectiveness. Overall, the rebuttal improves the clarity of the paper, but these points still affect my confidence in the strength of the claims.

---

> > > ### Author Response · Authors · 2026-04-04
> > >
> > > We sincerely thank the reviewer for the careful reading of our rebuttal and for the positive response. We appreciate the reviewer’s continued engagement and providing us the opportunity to further clarify these important points.
> > >
> > > ---
> > >
> > > ## **1. Completeness**
> > >
> > > We agree that in the context of Obliviate the notion of “true completeness” should be stated more carefully. Our intent is not to claim exact equivalence to full retraining but rather a principled approximation that closely matches the retrained solution up to bounded second‑order, curvature‑proxy, and low‑rank projection errors (Theorem H.14). We will revise the main text to explicitly qualify this claim and clarify that Obliviate provides strong, theoretically grounded practical completeness relative to retraining rather than exact completeness in a strict sense.
> > >
> > > ## **2. Local Convexity**
> > >
> > > We address the concern regarding the local strong convexity assumption as follows:
> > >
> > > ### **a. The Assumption is Local, Not Global**
> > >
> > > Our theoretical analysis requires strong convexity only in a *local neighborhood* around the trained solution $( \theta_0 )$, rather than globally across the entire parameter space. This is a significantly weaker and more realistic requirement. Specifically:
> > >
> > > * Even for non-convex models such as LightGCN, the loss landscape in the vicinity of a well-trained optimum often exhibits locally convex behavior.
> > > * This perspective is well-supported in prior literature, including: **[1]**, **[2]**, **[3]**.
> > >
> > > **[1]**  Pun, Y. M., & So, A. M. C. (2021). Local strong convexity of source localization and error bound for target tracking under time-of-arrival measurements. IEEE Transactions on Signal Processing, 70, 190-201.
> > > **[2]** Milne, T. (2019). Piecewise strong convexity of neural networks. Advances in neural information processing systems, 32
> > > **[3]** Jin, C., Ge, R., Netrapalli, P., Kakade, S. M., & Jordan, M. I. (2017, July). How to escape saddle points efficiently. In International conference on machine learning (pp. 1724-1732). PMLR.
> > >
> > > ### **b. Practical Justifications for Local Convexity**
> > >
> > > Several factors further support the plausibility of local convexity near $( \theta_0 )$:
> > >
> > > * **Regularization $( \Omega(\theta) )$**:
> > >   The L2 regularization term in the objective (Eq. 1) contributes a positive-definite component to the Hessian, encouraging local convexity.
> > >
> > > * **Adam’s curvature proxy**:
> > >   The diagonal preconditioner
> > >   $( \hat{H} = \mathrm{diag}(\sqrt{\hat{v}} + \epsilon) $
> > >   used in our method is positive-definite by construction, ensuring a well-conditioned curvature approximation regardless of the true Hessian structure (Section 2.1.3, Proposition F.2).
> > >
> > > * **Effectiveness of second-order approximation**:
> > >   The update
> > >   $( \theta^*_{-S} \approx \theta_0 - H^{-1} g_S )$
> > >   remains empirically effective, as discussed in Section 2.1.2.
> > >
> > > * **Two-stage robustness**:
> > >   Our design provides an additional safeguard: even if the Newton-style update (LUA) is imperfect due to mild non-convexity, the LAC stage corrects residual errors through direct optimization.
> > >
> > > * **Empirical validation**:
> > >   Obliviate achieves strong unlearning completeness and utility preservation with LightGCN, closely approximating retraining. Such performance would be unlikely if the local convexity assumption were severely violated.
> > > ---
> > >
> > > We thank the reviewer again for the thoughtful feedback, which has helped us improve both the clarity and precision of our claims. We hope that the above clarifications address the remaining concerns and strengthen confidence in the contributions of our work.

---

### Decision · Program_Chairs · 2026-04-30

**Decision:**

Accept (regular)

**Comment:**

The paper proposes a two-stage machine unlearning framework for recommender systems. The reviewers appreciated the use of Adam's second-moment statistics as a diagonal curvature proxy, and found the overall computational dividends of the proposed framework to be impressive. The theoretical analysis assumed local convexity, which the authors argued in the rebuttal that should be a reasonable approximation for a well-trained model (which should be at a local minimum), but is also encouraged by additional components of the studied setting (e.g., the ridge regularization term). A concern was raised also by the synthetic nature of removed interactions, but the authors handled it well, explaining the reasoning behind randomizing removals, but also the physical interpretation of various events that may indeed warrant removal (e.g., an errant click). Adding a quantitative completeness metric in the rebuttal strengthened the paper, but should have been considered from the beginning, and should be measured extensively across experiments. Additional experiments on demotion rates also strengthened the paper.

A remaining concern was comparing to recent baselines, as competitors were at least two years old. A reviewer pointed this out but failed to provide the relevant references, which unfortunately prevented the authors from performing a direct comparison during the rebuttal stage. The relevant references are:

[1] Post-Training Attribute Unlearning in Recommender Systems，ToIS24

[2] Plug and Play: Enabling Pluggable Attribute Unlearning in Recommender Systems, WWW25

but it is worth looking at papers that cite these two (relatively well-cited) works; potential competitors are numerous.

Computational efficiency is not the main focus of [1] and [2], and approaches are distinct from the one taken in the paper; that said, a direct comparison with recent, SoTA competitors would significantly strengthen the paper.

A nit: please cite the Adam paper correctly. Adam is the name of the method, not an author.